# ELFT: Efficient local-global fusion transformer for small object detection

**Guoguang Hua[1], Fangfang Wu[2], Guangzhao Hao[3], Chenbo Xia[2], Li Li [2]\***

**1** School of Artificial Intelligence, Guangzhou Maritime University, Guangzhou, Guangdong, China, **2** School of Information and Electrical Engineering, Hebei University of Engineering, Handan, Hebei, China, **3** Section of Network and Information, Handan Water Supply Co. Ltd, Handan, Hebei, China

\* hdlili@126.com

**Data availability statement:** All image files are available from the RSOD, NWPU VHR-10, and

## Abstract

Small object detection is an essential but challenging task in computer vision. Transformer-based algorithms have demonstrated remarkable performance in the domain of computer vision tasks. Nevertheless, they suffer from inadequate feature extraction for small objects. Additionally, they face difficulties in deployment on resource-constrained platforms due to their heavy computational burden. To tackle these problems, an efficient local-global fusion Transformer (ELFT) is proposed for small object detection, which is based on attention and grouping strategy. Specifically, we first design an efficient local-global fusion attention (ELGFA) mechanism to extract sufficient location features and integrate detailed information from feature maps, thereby promoting the accuracy. Besides, we present a grouped feature update module (GFUM) to reduce computational complexity by alternately updating high-level and low-level features within each group. Furthermore, the broadcast context module (CB) is introduced to obtain richer context information. It further enhances the ability to detect small objects. Extensive experiments conducted on three benchmarks, i.e. Remote Sensing Object Detection (RSOD), NWPU VHR-10 and PASCAL VOC2007, achieving 95.8%, 94.3% and 85.2% in mean average precision (mAP), respectively. Compared to DINO, the number of parameters is reduced by 10.4%, and the floating point operations (FLOPs) are reduced by 22.7%. The experimental results demonstrate the efficacy of ELFT in small object detection tasks, while maintaining an attractive level of computational complexity.

## 1 Introduction

Object detection is an active topic in computer vision, with extensive applications in face detection, autonomous driving, and security monitoring. In particular, small object detection is a vital area of concern. It holds crucial application value across various fields, such as aerial image analysis, satellite remote sensing, medical image interpretation, and industrial automation. However, small object detection faces many challenges, such as a small proportion of pixels, insufficient semantic information, and occlusion in complex scenes. These challenges make it more difficult to detect small objects accurately. Additionally, deep

PASCAL VOC2007 database (accession number(s) https://github.com/RSIA-LIESMARS-WHU/RSOD-Dataset, : https://github.com/Wufang2/NWPU_VHR-10, http://host.robots.ox.ac.uk/pascal/VOC/voc2007/index.html).

**Funding:** The work was funded by Science and Technology Research and Development Plan Project of Handan, Hebei Province, China (21422031289) and Science Research Project of Hebei Education Department, China (SQ2023096). The funders had no role in study design, data collection and analysis, decision to publish, or preparation of the manuscript.

neural networks for small object detection tend to degrade the spatial information as it traverses through the layers of the network. The global dependency feature of the Transformer [1] can mitigate this problem. However, Transformer-based methods encounter problems including substantial computational requirements and slow training processes. These issues hinder the ability to meet the demands of real-time detection and platforms with limited resources. Some approaches design networks via the integration of local and global strategies, aimed at extracting both global and local features. For example, the Fourier Neural Operator with Local Priors and Global Perceptron (FOLGP) proposed in [2] integrates Transformer modules to model global contextual information across frequency bands, enhancing the capture of correlations between different frequency components in the pantograph-catenary system (PCS). This integrated approach stands in contrast to other strategies: local feature extraction methods like the self-supervised pre-training method [3] employ domain-specific masking to retain local structural features of railway components, achieving high efficiency in small object detection. Meanwhile, multi-modal methods such as [4] fuse local cues from heterogeneous sensors (e.g., infrared/visible) to resolve ambiguities in complex scenes. However, these methods lack explicit mechanisms for modeling global system dynamics. Therefore, it is still an open question to achieve efficient and accurate detection of small objects.

The rise of convolutional neural networks has brought small object detection into the era of deep learning, significantly improving the intelligence and automation levels of small object detection. The improved Faster R-CNN [5] enhanced small object detection by adopting Improved IoU loss for bounding box regression, bilinear-interpolated RoI pooling to reduce localization errors, and multi-scale feature fusion to strengthen representation capability. Building on this, the enhanced SSD [6] introduced feature cross-reinforcement, an improved group shuffling-efficient channel attention mechanism and an adaptive training sample selection algorithm to address challenges of small object detection. Specifically for UAV aerial imagery, Chen et al. [7] proposed SOD-YOLOv7, which integrated Swin Transformer with a Bi-Level Routing Attention (BRA) mechanism within its feature extraction network. Along with the advancement of Transformer in natural language processing tasks, the Facebook team introduced the Detection Transformer (DETR) [8] object detection algorithm. DETR transformed the task of object detection into a set prediction problem, opening up a novel approach to the field. Even though DETR eliminated manual processing steps such as anchor box design and Non-Maximum Suppression in traditional object detection methods, it still has evident drawbacks like slow convergence rate, low detection accuracy for small objects, and high computational resource requirements. DINO [9] introduced contrastive denoising training on the basis of Deformable DETR [10], DAB-DETR [11], and DN-DETR [12]. The innovative method accelerated the convergence rate during training and enhanced the stability of the algorithm, laying the foundation for future research. However, it leaves much room for enhancing its detection capabilities, particularly when it comes to identifying small objects. In the task, we need to simultaneously solve the problems of insufficient feature information and large computational cost, which make it more challenging.

To address this challenge, we explore the DINO-based object detection algorithm and propose an efficient local-global fusion Transformer (ELFT) for small object detection. ELFT integrates three novel components, i.e., efficient local-global fusion attention module (ELGFA), grouping feature update module (GFUM) and context broadcast module (CB). Concretely, the ELGFA is designed to ensure the accurate location information for the region of interest, which independently process feature vectors in global, vertical, and horizontal directions. We further propose GFUM to increase encoder efficiency and reduce computational complexity by alternately updating high-level and low-level features. As for the CB module, it is employed to obtain global context and enhance the precision of detecting small

objects. In summary, ELFT leverages attention mechanisms and grouping strategy for the trade-off between performance and computational complexity. The main contributions are summarized below:

1. ELFT is proposed to address the limitations of feature extraction networks in capturing long-range dependencies. Furthermore, it aims to mitigate the high computational complexity that is inherent in Transformer-based detection methods.
2. We introduce an efficient local-global fusion attention, a context broadcast module and a grouping feature update module to achieve a fine balance between high performance and reasonable computational cost.
3. Experiments are conducted on three public benchmarks: RSOD, NWPU VHR-10, and PASCAL VOC2007, and the results show that the proposed method improved detection accuracy while reducing parameters and computational complexity.

The remainder sections of this article are structured as outlined below. The "Related Work" section provides an overview of relevant literature. "Efficient Local-Global Fusion Transformer Algorithm" delves into the specifics of our proposed ELFT architecture. "Experiments and Results" section showcases the experimental results and their corresponding analysis. Lastly, the "Conclusion" section summarizes the key points discussed in this article.

## 2 Related works

In this part, we offer a brief overview of related research on small object detection methods based on DETR, lightweight methods based on DETR, and attention mechanisms.

### 2.1 Small object detection methods based on DETR

Effectively representing objects at diverse scales has remained a core challenge in the object detection field. Especially for small-scale objects, which occupy limited pixel space in an image, carrying relatively little feature information. Therefore, it is crucial to explore and utilize the detailed information in the image fully. Table 1 demonstrates the studies related to small object detection based on DETR. From the multi-scale features perspective, Zhu et al. [10] improved the accuracy of detecting small objects by adopting a multi-scale feature fusion strategy to merge information from different scales. On this basis, Cunha et al. [13] replaced the simple data augmentation in Deformable DETR with a simple augmentation technology for AUGMIX classification for adapting to small objects. To address the weak correlation between network layers in DETR and the poor performance in small object training and detection. Skip DETR [14] enhanced feature fusion through skip connections and multi-scale feature extraction to provide an efficient solution for forestry pest detection. In contrast, Cao et al. [15] designed a decoder layer structure that progressed from coarse to fine, approaching the problem from the perspective of bounding box localization. The structure incorporated an adaptive scale fusion module to merge feature information from different scales using object query features, thus refining small object localization. Meanwhile, Hoanh et al. [16] introduced an object-focus network with a dual-head, where one head served as a dedicated small object prediction module for obtaining coarse locations of small objects. From the perspective of the limitations of inductive bias, Dubey et al. [17] addressed DETR's shortcomings in small object detection by feature fusion and normalized inductive bias design, while preserving the end-to-end advantage of ensemble-based prediction.

As analyzed above, these researchers primarily focused on multi-scale features, data augmentation techniques, coarse location acquisition and inductive bias design. However, the

**Table 1. The studies related to small object detection based on DETR.**

| Model | Pros | Cons | Originality | Datasets | Results |
|---|---|---|---|---|---|
| Deformable DETR [10] | a) Enhanced small-object detection. b) Accelerated convergence. | High computational complexity. | a) Deformable attention mechanism focusing on key sampling points. b) Iterative bounding box refinement. | COCO 2017 | FLOPs=173G Params=40M mAP=46.2% |
| DETRAug [13] | a) Robust to occlusion and low-resolution inputs. b) Adaptable to small-object detection. | Increased training time due to augmentation. | Augmentation strategy inspired by AUGMIX classification. | detect-waste | Params=39.8M mAP=48.0% |
| Skip DETR [14] | a) Improved small-object detection. b) Accelerated convergence. | Precision-computation balance challenges. | a) Skip connections for feature enhancement. b) Spatial pyramid pooling for query initialization. | Forestry Pest | FLOPs=5.14G Params=36.8M mAP=74.1 |
| CF-DETR [15] | a) Improved localization accuracy for small objects. b) Addressed DETR's slow convergence. | Increased memory and computational requirements. | a) Coarse-to-fine decoder with local-global feature fusion. b) Transformer-enhanced FPN for multi-scale representation. | COCO 2017 | mAP=47.8% |
| Hoanh and Pham [16] | High detection accuracy of small objects in dense high-resolution images. | Higher model complexity and resource requirement. | a) Dual-head architecture design. b) Feature pyramid fusion with Transformer. | VisDrone COCO 2017 | FLOPs=164G mAP=34.9% FLOPs=169G mAP=47.8% |
| SOF-DETR [17] | a) Improved small-object features. b) Global end-to-end context preservation. | a) Limited generalization. b) High resource consumption. | a) Normalized inductive bias in Transformer for small-object representation. b) Lazy feature fusion for contextual information. | COCO 2017 | mAP=42.7% |

detection performance for small objects remains inadequate due to their low pixel ratio and frequent occlusion. Consequently, fully exploring and leveraging the feature information of small objects to enhance detection accuracy remains a critical research direction. In this paper, we strengthen the precision of detecting small objects by combining local and global contextual information, which is a feasible and efficient solution.

## 2.2 Lightweight methods based on DETR

The growing demand for object detection in practical applications, especially on resource-constrained platforms. It has made lightweight object detection algorithms become an urgent topic and research hotspot in both academia and industry. Table 2 demonstrates the lightweight methods based on DETR. Sparse DETR [18] utilized the decoder's cross-attention map (DAM) to guide feature selection, and filtered encoder outputs with learnable cross-attention to reduce computational cost. However, not all the feature vectors filtered by Sparse DETR corresponded to foreground regions. Based on this observation, Huawei Noah's Ark Lab supposed that Sparse DETR used DAM to monitor the foreground feature vectors, but DAM could introduce errors during training. Therefore, Focus-DETR [19] was proposed to better monitor the filtering process of foreground feature vectors by utilizing actual boxes and labels. It achieved better performance by introducing positioning and semantic information for multi-level semantic discrimination. Sun et al. [20] proposed Pruning DETR, which adjusted the importance of module outputs by leveraging parameter scale factor and a sparse regularization term. It pruned the DETR structurally, resulting in a decrease in computational burden and improved inference speed. Furthermore, Zheng et al. [21] introduced an efficient attention mechanism to speed up Transformer model, which was combined linear attention and token pruning.

The above methods reduced the computational load by filtering feature vectors, decreasing attention complexity, and pruning. However, they exhibited notable drawbacks in terms

**Table 2**. The lightweight methods based on DETR.

| Model | Pros | Cons | Originality | Datasets | Results |
|---|---|---|---|---|---|
| Sparse DETR [18] | a) Reduced computational cost. b) Maintained detection performance. | a) Complex training pipeline. b) Limited robustness in complex scenarios. | Encoder token sparsification via objectness score and decoder attention maps. | COCO 2017 | FLOPs=136G mAP=46.3% |
| Focus-DETR [19] | Balanced computation efficiency and accuracy. | The computational complexity is still high. | Multi-level semantic discrimination with position-aware scoring. | COCO 2017 | FLOPs=154G Params=48M mAP=50.4% |
| Pruning DETR [20] | Reduced inference latency and memory usage. | a) Performance drop. b) Pruning-induced robustness issues. | Structured pruning with sparsity regularization and parameter scaling. | COCO 2017 | FLOPs=126G Params=26M mAP=42.7% |
| Zheng et al. [21] | a) Reduced computational complexity. b) Enhanced deployment flexibility. | Accuracy degradation. | Linear attention and token pruning for acceleration. | ImageNet COCO 2017 | The FLOPs reduced by 30%–50% Params=39M |

of detection accuracy. Thanks to the grouped feature update module, our algorithm improves the encoder's efficiency and reduces computational complexity while maintaining the detection accuracy.

## 2.3 Attention mechanisms

The attention mechanisms simulate the human visual and cognitive processes, enabling the neural network to autonomously learn and dynamically concentrate on the critical parts of the input data, ultimately improving the algorithm's effectiveness. The main structure of SENet [22] was the Sequence-and-Exception (SE) block. By incorporating the attention mechanism into the image channel, the algorithm paid more attention to the channel features containing a large amount of information. To exploit the spatial correlation details of the image, a model that analyzes pairwise pixel correlations and hierarchical statistics is established [23]. DANet [24] adopted both spatial and channel attention mechanisms to extract the spatial and channel information in the image, which improved the accuracy of segmentation. Even though they boosted the performance, most of them ignored the width and height information of the object. To alleviate this issue, Zhang et al. [25] designed a Quadruple Attention module to extend the attention mechanism from distinct dimensions of channels, positions, heights, and widths, thus better representing the characteristics of small objects. The Magnifying Glass module further emphasized small objects during detection and further improved its detection performance. Multi-head Self-attention (MSA) enabled the model to learn diverse data representations within different subspaces, thereby obtaining various information. Hyeon-Woo et al. [26] conjectured that MSA tended to learn intensive interactions, but the gradient of intensive self-attention was steeper, leading to unstable training. Hence, they proposed the Context Broadcast module (CB) and dimension-scaled CBs to provide uniform attention, which effectively improved the performance of ViT. For the purpose of paying attention to the lesions of COVID-19, Zhou et al. [27] designed SE-Res block by adding the residual connection, and subsequently introduced it to ResNet. Srinivas et al. [28] integrated the self-attention mechanism into ResNet. The image was processed using convolutions to obtain localized information. Subsequently, the global dependency was modeled utilizing the self-attention mechanism. This approach surpassed the baseline ResNet model in object detection performance..

As claimed above, the attention mechanism had been successfully integrated into ResNet. However, two major issues arose: on the one hand, the lack of long-range dependencies restricts the algorithm's capacity to detect small objects, compromising both accuracy and

reliability. On the other hand, its non-lightweight design did not apply to the devices with constrained computing resources more challenging. To overcome such problems, this paper designs a lightweight ELGFA module with a simple structure and few parameters, which can effectively capture the long-range spatial dependencies and accurately locate the location of the target object.

## 3 Efficient local-global fusion transformer algorithm

In this part, we introduce the network architecture of ELFT. Subsequently, we offer a comprehensive explanation regarding the proposed ELGFA module, along with the GFUM, and CB module.

### 3.1 Network architecture

The Transformer model captures global dependencies within a sequence through its attention mechanism, which is primarily composed of encoders and decoders. The encoder consists of multiple stacked self-attention layers, followed by a Feed-Forward Network (FFN) layer. The principle of the self-attention mechanism lies in calculating the relationship among every element in the sequence relating to the others. These calculations are described as Eq (1). The FFN layer consists of two linear transformations separated by an activation function.

$$Attention(Q, K, V) = softmax\left(\frac{QK^T}{\sqrt{d_k}}\right)V \tag{1}$$

For the input sequence $X$, the query matrix $Q$, the key matrix $K$, and the value matrix $V$ are obtained through linear transformations. Then, the attention scores are calculated using the dot product of $Q$ and $K$. A scaling factor $\sqrt{d_k}$ is applied to mitigate the potential vanishing gradient issue in the softmax operation caused by large dot product values. The attention weights are computed by applying the softmax function to the attention scores. Ultimately, the computation of the final output involves calculating the weighted total of the $V$, where the weights are determined by the attention weights.

The structure of the decoder resembles that of the encoder, but it has a cross-attention layer, which takes into account the output of the encoder. The definition of cross-attention is given by Eq (2):

$$Attention(Q, K, V) = softmax\left(\frac{Q(K^T)}{\sqrt{d_k}}\right)V \tag{2}$$

where $K$ and $V$ represent the outputs from the encoder, while $Q$ stems from the input of the decoder.

DETR primarily consists of a backbone network, Transformer encoders, decoders, and FFN structure. DINO is an improved method based on DETR. The specific detection process is depicted in Fig 1. When an image is received, multi-scale features are extracted by backbone networks, for instance ResNet or Swin-Transformer. These features are then input into the Transformer encoder alongside their respective position embeddings. After encoding and enhancing the input features, the encoder initializes the decoder's location queries by associating them with the positional information of the selected top-k features. The Transformer decoder receives both these queries and the content queries to assist in obtaining better location information. The decoder then uses deformable attention to integrate the encoder's feature outputs and iteratively updates the query layer on a layer-by-layer basis. The predicted

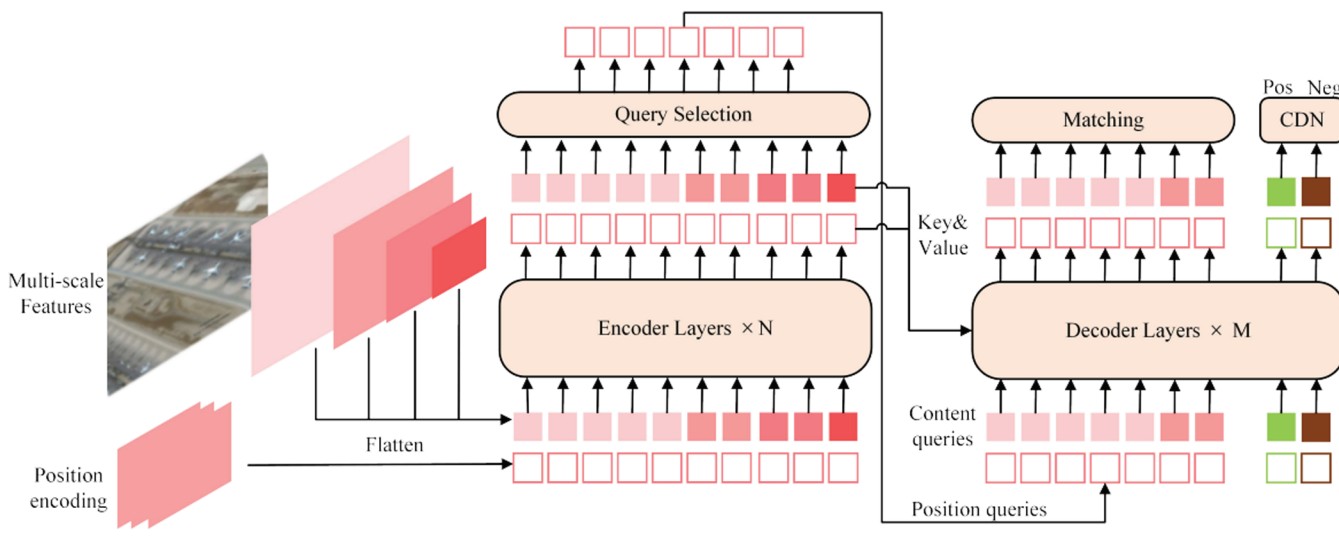

**Fig 1. The network architecture diagram of DINO.**

bounding box generated by the decoder is matched with the actual object box. The Hungarian matching process involves calculating category loss, positioning loss, and GIoU loss separately after obtaining the predictions, along with the actual labels and boxes. These losses are then combined to form a cost matrix, which is used to compute the matching labels and prediction boxes. Concurrently, there is an additional branch for contrast denoising training. Each actual box generates a positive sample and a negative sample. The label is augmented with noise and appended to the actual box. The boxes with noise are marked as positive, while the others are negative, to avoid the same object being predicted again. Finally, the ultimate prediction box is determined jointly by the initial box and the predicted offset. To mitigate the computational complexity and enhance the detection capability for small objects, we propose an advanced DINO algorithm, named ELFT, as illustrated in Fig 2. To reduce overall computational demands, we first optimize the encoder and backbone network. GFUM is designed to alternately refresh both high-level and low-level features, thereby enhancing the encoder's efficiency and minimizing the algorithm's computational cost. During image processing, the backbone network ResNet50 suffers from insufficient context information acquisition. The loss of spatial information through multi-layer network transmission often results in missed and false detections of small objects. In addition, the network structure of ResNet50 is relatively complex. This will lead to the need for more computing resources and time during training. To circumvent this, the ELGFA module is designed to replace the last two bottlenecks of ResNet50, thus diminishing the parameter count and computational cost, while enhancing detection precision. Additionally, the multi-head self-attention mechanism tends to learn intensive interactions. When the attention weights are very close, this intensive self-attention gradient is steeper, and the training process is more complicated. For the purpose, the CB module is introduced to provide uniform attention and help stabilize the training. By broadcasting the global contextual data, it ensures that the entire input data is

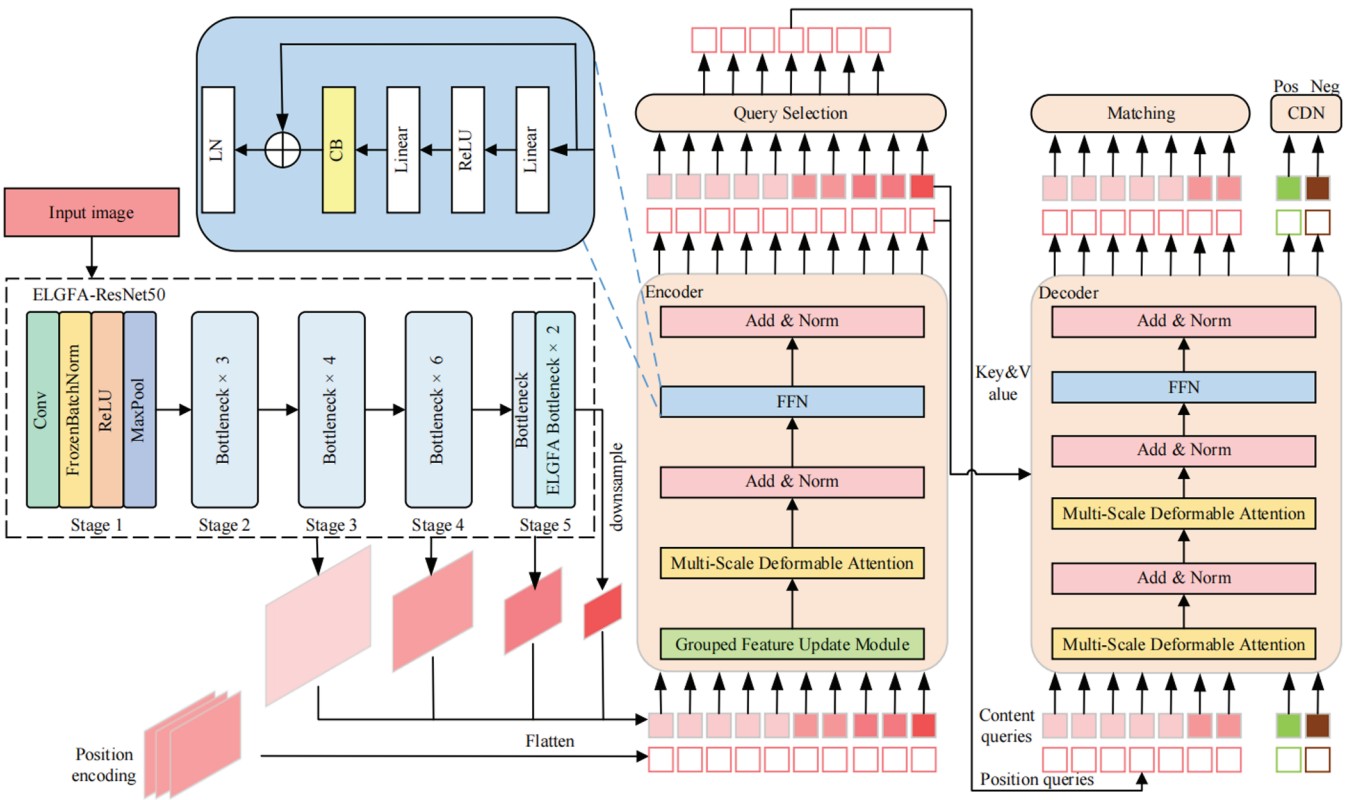

**Fig 2. The architecture of ELFT.**

considered comprehensively during the processing of each position. This promotes to distinguish small objects from background information or other objects. The pseudo-code of the overall method is shown in Algorithm 1. In contrast to DINO, ELFT achieves a reduction in both parameters and computational demands, and substantially boosts the effectiveness in detecting small objects.

## 3.2 Efficient local-global fusion attention

For the small object detection task, using ResNet50 as the backbone network for image feature extraction often leads to insufficient context information. As the network depth increases, the resolution and detailed information of the features tend to diminish, further compromising context acquisition. This results in an increased likelihood of false and missed detections of small objects. Moreover, the network structure of ResNet50 is relatively complex and has a large number of parameters, which will cause an urgent demand for more computing resources and time during training. Motivated by Efficient Local Attention [29], we introduce ELGFA. By pooling image features from the height, width, and global dimensions, ELGFA effectively integrates both local and global contextual information to enhance the input feature image, aiming to accurately identify regions of interest. The structure of ELGFA module is presented in Fig 3, and its processing flow is as follows: Specifically, the input feature map $x$ undergoes a global average pooling process, yielding a tensor of the dimensions (batch_size,

**Algorithm 1 ELFT.**

**Require:** set of input RGB image $I$

**Ensure:** Bounding boxes $B$ and class labels $C$

1: Backbone = ELGFA-ResNet50
2: Features ← Backbone($I$)
3: Encoder = TransformerEncoder with GFUM and CB(layers = 6)
4: EncodedFeatures ← Encoder(Features, PositionalEmbeddings)
5: TopKFeatures ← SelectTopK(EncodedFeatures, $K$ = 900)          ▷ Based on
    objectness score
6: PositionalQueries ← TopKFeatures.positions     ▷ Dynamic anchor boxes
7: ContentQueries ← LearnableParameters          ▷ Static content queries
8: Decoder = TransformerDecoder() with CB
9: **for** layer in Decoder **do**
        PositiveQueries ← AddNoise(GT_boxes, scale=$\lambda_1$)
        NegativeQueries ← AddNoise(GT_boxes, scale=$\lambda_2$)          ▷ $\lambda_1 < \lambda_2$
        CDNQueries ← Concat(PositiveQueries, NegativeQueries)
10: **end for**
        RefinedQueries ← DecoderLayer(CurrentQueries, EncodedFeatures)
        Predictions ← PredictionHeads(RefinedQueries)
        $B$ ← Predictions.boxes
        $C$ ← Predictions.classes
11: **return** $B$, and $C$

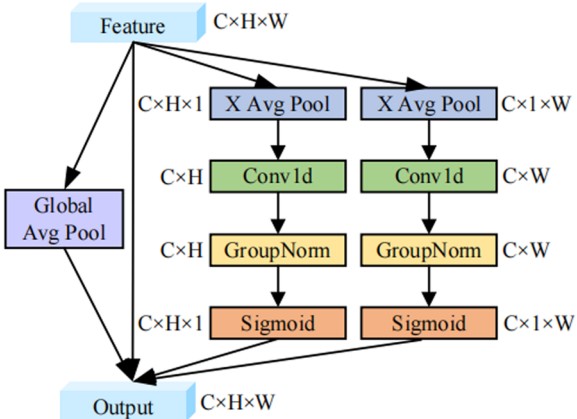

**Fig 3. The structure of ELGFA module.**

channels, 1, 1), which contains the global context information $gc$ for each channel. The estimation of global context is expressed as Eq (3):

$$gc(x)_{b,c} = \frac{1}{h \times w} \sum_{i=1}^{h} \sum_{j=1}^{w} x_{b,c,i,j} \tag{3}$$

where $x$ denotes the input feature map of size (b, c, h, w), where $b$, $c$, $h$, $w$ refer to the batch_size, channels, height, and width of the feature map, respectively. $gc(x)_{b,c}$ is the result of global average pooling for channel $c$ in sample $b$.

After that, the input feature map $x$ performs average pooling along the width and height directions, and the tensors $p_c^h(h)$ and $p_c^w(w)$ are obtained. Here, $p_c^h(h)$ represents the local context along the height direction $(H,1)$, with dimensions (b, c, h). Similarly, $p_c^w(w)$ denotes the local context along the width direction $(1, W)$, with dimensions (b, c, w). They are calculated by (Eqs 4) and (5), respectively.

$$p_c^h(h) = \frac{1}{H} \sum_{0 \leq i \leq H} x_c(h, i) \tag{4}$$

$$p_c^w(w) = \frac{1}{W} \sum_{0 \leq j \leq W} x_c(j, w) \tag{5}$$

The one-dimensional convolution with a convolution kernel of 7 is employed to effectively extract features while enhancing the horizontal and vertical positioning information. Then, the feature vectors from vertical and horizontal directions perform group normalization and nonlinear activation functions to generate bidirectional positional attention predictions. The whole processes are defined as Eqs (6) and (7):

$$y^h = \sigma(G_n(F_h(p_h))) \tag{6}$$

$$y^w = \sigma(G_n(F_w(p_w))) \tag{7}$$

where $p_h$ and $p_w$ are the results of Eqs (4) and (5), respectively. $F_h$ and $F_w$ refer to 1D convolutions in vertical and horizontal directions, respectively. $G_n$ represents group normalization and $\sigma$ indexes the nonlinear activation function Sigmoid.

The weight of each position in the final feature map is determined by a combination of global context information and attention weights in both vertical and horizontal directions. The original input $x$ performs element-by-element multiplication with the enhanced local contexts $y^h$ and $y^w$, along with the global context $gc$. This approach allows the algorithm to consider both local and global context information, thereby enhancing the feature representation. As shown in Eq (8):

$$Y = x_c \times y^h \times y^w \times gc \tag{8}$$

ResNet50 mitigates the vanishing gradient problem in deep networks through residual learning, thereby enabling deeper architectures. Its hierarchical feature extraction structure generates multi-scale features that are crucial for object detection. By leveraging pre-trained weights, the network achieves rapid convergence through optimal initialization. This residual mechanism strikes a balance between network depth and computational efficiency, ultimately achieving a trade-off between performance and overhead. Therefore, adopting ResNet50 as the backbone network represents an appropriate choice. However, a key limitation of ResNet50 lies in its local convolutional operations, which fail to capture distant contextual dependencies critical for small object detection. In this paper, we integrate the proposed ELGFA module into ResNet50 to capture global context information. Specifically, the convolution layer within the bottleneck of the last two layers of the S5 block is modified to form ELGFA-ResNet50 (see Table 3). It not only decreases the computational requirements of the feature extraction network, but also enhances the precision of detection.

**Table 3. Structure of comparison between ResNet50 and ELGFA-ResNet50.**

| Backbone | output | ResNet-50 | ELGFA-ResNet50 |
|---|---|---|---|
| S1 | 512×512 | 7×7,64,stride 2 | 7×7,64,stride 2 |
| S2 | 256×256 | 3×3 max pool,stride 2 $\begin{bmatrix} 1\times1,64 \\ 3\times3,64 \\ 1\times1,256 \end{bmatrix} \times3$ | 3×3 max pool,stride 2 $\begin{bmatrix} 1\times1,64 \\ 3\times3,64 \\ 1\times1,256 \end{bmatrix} \times3$ |
| S3 | 128×128 | $\begin{bmatrix} 1\times1,128 \\ 3\times3,128 \\ 1\times1,512 \end{bmatrix} \times4$ | $\begin{bmatrix} 1\times1,128 \\ 3\times3,128 \\ 1\times1,512 \end{bmatrix} \times4$ |
| S4 | 64×64 | $\begin{bmatrix} 1\times1,256 \\ 3\times3,256 \\ 1\times1,1024 \end{bmatrix} \times6$ | $\begin{bmatrix} 1\times1,256 \\ 3\times3,256 \\ 1\times1,1024 \end{bmatrix} \times6$ |
| S5 | 32×32 | $\begin{bmatrix} 1\times1,512 \\ 3\times3,512 \\ 1\times1,2048 \end{bmatrix} \times3$ | $\begin{bmatrix} 1\times1,512 \\ 3\times3,512 \\ 1\times1,2048 \end{bmatrix} \times1$ $\begin{bmatrix} 1\times1,512 \\ ELGFA,512 \\ 1\times1,2048 \end{bmatrix} \times2$ |

## 3.3 Grouped feature update module

The computational cost of the encoder in DINO accounts for 58.3% of the entire algorithm. Among these calculations, high-level features contribute only one-quarter, while low-level features account for three-quarters of all tokens processed. This is one of the reasons for the high computational cost of the encoder [30]. To deal with this issue, we propose GFUM and integrate it into the encoder to alternately update high-level and low-level features. The architecture of GFUM is displayed in Fig 4, and the pseudo code for alternate update process is presented in Algorithm 2. To be more specific, the six encoder layers are organized into three groups, with each group comprising two encoder layers. In the first encoder layer of each group, high-level features are used to query all tokens, updating the feature vector while low-level features remain unchanged. In this way, the number of queried features shrinks to one-fourth of the original, thus decreasing the overall computational burden. In the second encoder layer of each group, low-level features serve as queries to interrogate all tokens,

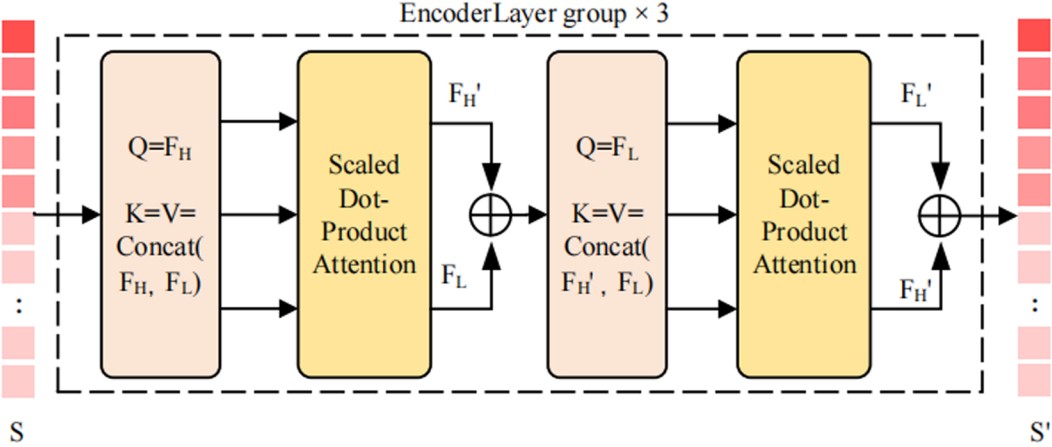

**Fig 4. The structure of GFUM.**

enabling an update of their respective representations while preserving the integrity of multi-scale features. Through the grouping and updating alternately method, efficient computation is achieved.

**Algorithm 2 Alternately update algorithm.**

**Require:** set of input feature maps *src*, set of target feature maps *tgt*, total number of layers *layers*, number of groups *num_groups*, position encoding *pos*, reference points *ref_points*

**Ensure:** set of processed feature maps *src*

1: **for** $layer\_id \leftarrow 0$ **to** $layers - 1$ **do**
2: **if** $(layer\_id + 1)\%(layers/num\_groups) == 0$ **then**
3: $tgt \leftarrow src$
4: $output \leftarrow \text{LAYER}(src = src, tgt = tgt, pos = pos\_input, ref\_points = ref\_points\_input)$
5: $tgt \leftarrow output[:, \texttt{level\_start\_index}[4 - enc\_scale]:]$
6: $src \leftarrow output$
7: **else**
8: $output \leftarrow \text{LAYER}(src = src, tgt = tgt, pos = pos\_tgt, ref\_points = ref\_points\_tgt)$
9: $tgt \leftarrow output$
10: $src \leftarrow \texttt{concatenate}([src[:, :\texttt{level\_start\_index}[4 - enc\_scale]], tgt], 1)$
11: **end if**
12: **end for**
13: **return** *src*

## 3.4 Context broadcasting module

The multi-scale deformable attention module in the Transformer architecture serves as the core component to capture information from every position within an image. It facilitates intensive interactions by increasing the diversity of attention mechanisms. Each attention head independently learns a unique attention weight distribution to capture information from different areas in the image. However, the diversity is prone to a more complex distribution of attention weights, where each position may receive a relatively high attention weight. This potentially diminishing the learning efficiency of the algorithm.

To further improve the algorithm's capability of perceiving global context information, we introduce a CB module into the FFN after the multi-scale deformable attention module, as shown in Fig 5. The CB module computes the average of feature vectors across layers and then rebroadcasts this global context information to each feature vector. In this manner, each feature vector can acquire a context representation based on the average information from all feature vectors. The introduction of CB module enables the algorithm to learn more intensive interactions. Each vector considers not only its local context but also the global context. It is worth noting that CB improves the algorithm's ability to accurately locate and identify small objects. In the mean time, the introduction of global information contributes to the stable training process of the algorithm.

## 4 Experiments and results

In this part, we conduct experiments on the RSOD, NWPU VHR-10, and PASCAL VOC datasets. Furthermore, we evaluate ELFT by comparing it with state-of-the-art methods. Additionally, we carry out ablation studies to assess the efficacy of the three modules we have introduced in ELFT.

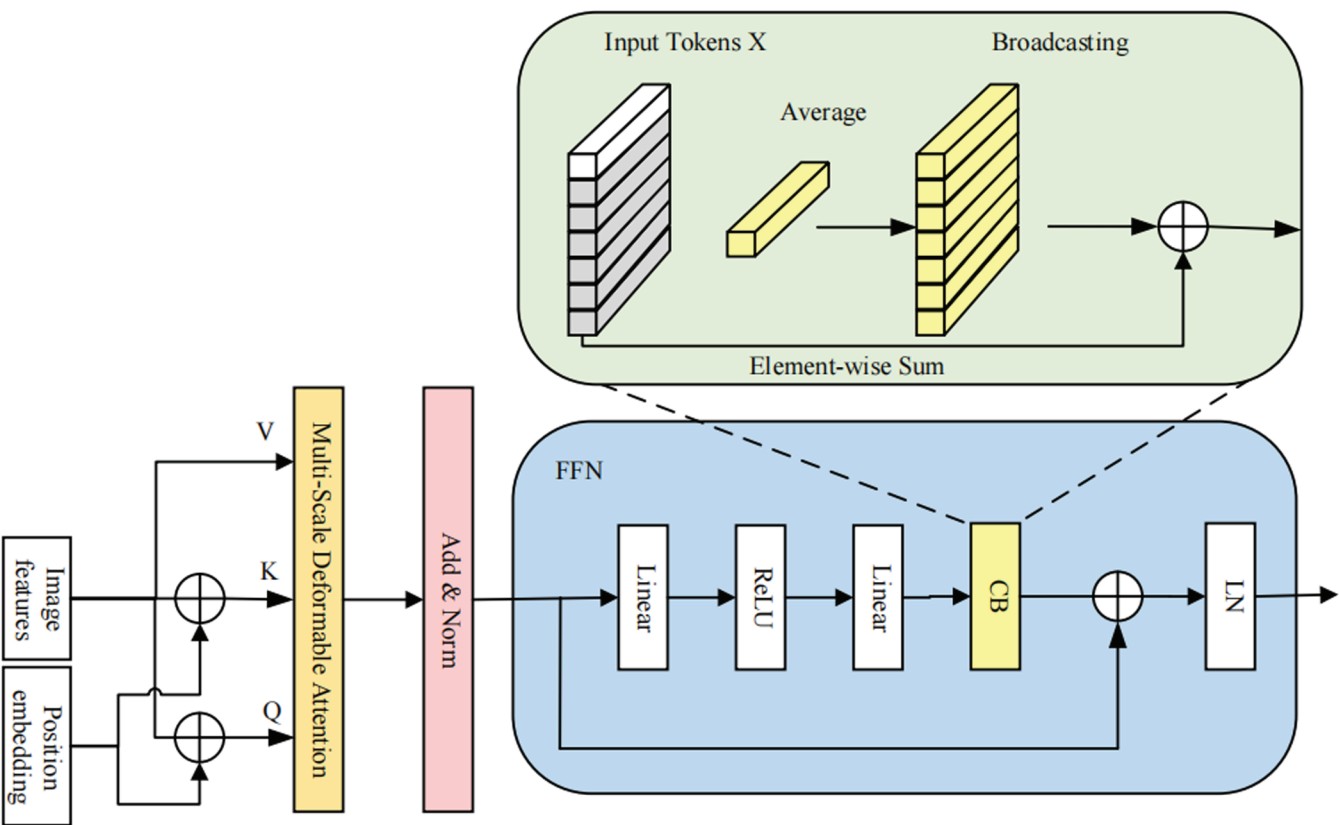

**Fig 5. The structure of CB module in FFN.**

## 4.1 Experiment details

In this paper, we implement ELFT based on the deep learning framework Pytorch. All experiments are conducted on 12th Gen Intel Core i9-12900H 2.50 GHz and NVIDIA GeForce RTX 3060 GPU. The implementation utilizes Python as the programming language, with Windows 11 as the operating system. During training, AdamW [31] is employed to optimize the network with a batch size of 1, weight decay of 0.0001, and a training period of 12 epochs. The learning rate is initialized to 0.00001, and is decreased by a factor of 10 after 11 epochs.

## 4.2 Datasets

The RSOD dataset [32,33], which consists of 976 images, is an openly accessible resource specifically designed for object detection within remote sensing imagery. The dataset contains four different object categories: aircraft, oil tank, overpass, and playground, with a total of 6,950 objects. Among them, 4,993 aircraft objects are marked in 466 images, 1,585 oil tank objects in 165 images, 180 overpass objects in 176 images, and 191 playground objects in 189 images. This paper divides the RSOD dataset into a training set and a validation set, with a ratio of 70% allocated to the training set and 30% to the validation set.

NWPU VHR-10 dataset [34–36] is a public remote sensing dataset released by Northwestern Polytechnic University. It contains 800 pictures, belonging to 10 types of objects, 650 pictures with objects, and 150 pictures with backgrounds. The dataset has been partitioned into a training set and a validation set in a 7:3 ratio.

PASCAL VOC2007 dataset [37,38] contains 9,963 images from 20 categories, which are divided into the train-val2007 dataset and the test2007 dataset. In this paper, train-val2007 is used as a training set, including 5,011 images, and the model is evaluated on the test2007 dataset, including 4,952 images.

### 4.3 Metrics

We employ the average precision (AP), mean average precision (mAP), floating point operations (FLOPs), and F1-score, and Optimal Localization Recall Precision (oLRP) [39] as the evaluation metrics to assess the performance of the algorithm. The *AP* denotes the area surrounded by the accuracy curve *Precision*, recall curve *Recall*, and coordinate axis. The *Precision* refers to the fraction of accurately predicted positive samples out of the total number of samples predicted to be positive. The *Recall* represents the fraction of positive samples that are correctly predicted among all actual positive samples. Specifically, they can be defined by Eqs (9) and (10):

$$Precision = \frac{TP}{TP + FP} \tag{9}$$

$$Recall = \frac{TP}{TP + FN} \tag{10}$$

where *TP* denotes the positive samples that are correctly predicted, *FP* denotes the negative samples that are incorrectly predicted as positive, and *FN* represents the positive samples that are incorrectly predicted as negative. *AP* can be calculated by Eq (11).

$$AP = \int_0^1 P(R)dR \tag{11}$$

*mAP* is the average of different kinds of *AP*. *mAP* can be calculated by Eq (12) as follows:

$$mAP = \frac{AP_1 + AP_2 + \cdots + AP_n}{n} \tag{12}$$

where *n* is the number of object categories.

The F1-score serves as the harmonic average of Precision and Recall. A higher F1-score signifies an optimal trade-off between Precision and Recall, reflecting the algorithm's performance. F1-score is estimated as Eq (13).

$$F1 = 2 \times \frac{Precision \times Recall}{Precision + Recall} \tag{13}$$

FLOPs are employed as a metric to quantify the computational demand and complexity of the algorithm. A larger FLOPs value signifies a higher requirement for computing resources.

oLRP comprehensively evaluates localization accuracy, precision error, and recall error. Under an IoU threshold, it calculates the tightness of bounding boxes enclosing the object to obtain a more reliable assessment of localization performance. Let *X* represent the set of ground-truth boxes and *Y* denote the detector's predicted box set. When given a confidence score $s(0 \leq s \leq 1)$ and an IoU threshold $\tau(0 \leq \tau < 1)$, the *LRP* error of $Y_s$ relative to *X* can be calculated by Eq (14):

$$LRP(X, Y_s) = \frac{1}{Z}(w_{IoU}LRP_{IoU}(X, Y_s) + w_{FP}LRP_{FP}(X, Y_s) + w_{FN}LRP_{FN}(X, Y_s)) \tag{14}$$

where $Z = N_{TP} + N_{FP} + N_{FN}$ denotes the number of *TP*, *FP* and *FN* samples. $w_{IoU} = \frac{N_{TP}}{1-\tau}$, $w_{FP} = |Y_s|$, $w_{FN} = |X|$ represent weights of components. $LRP_{IoU}$, $LRP_{FP}$, and $LRP_{FN}$ can be defined by Eqs (15), (16) and (17):

$$LRP_{IoU}(X, Y_s) = \frac{1}{N_{TP}} \sum_{i=1}^{N_{TP}} (1 - IoU(x_i, y_{x_i})) \qquad (15)$$

$$LRP_{FP}(X, Y_s) = 1 - Precision = 1 - \frac{N_{TP}}{|Y_s|} = \frac{N_{FP}}{|Y_s|} \qquad (16)$$

$$LRP_{FN}(X, Y_s) = 1 - Recall = 1 - \frac{N_{TP}}{|X|} = \frac{N_{FN}}{|X|} \qquad (17)$$

In summary, the *LRP* error can be organized as Eq (18).

$$LRP(X, Y_s) = \frac{1}{N_{TP} + N_{FP} + N_{FN}} \left( \sum_{i=1}^{N_{TP}} \frac{1 - IoU(x_i, y_i)}{1 - \tau} + N_{FP} + N_{FN} \right) \qquad (18)$$

The *oLRP* is defined as the minimum *LRP* error achievable at $t = 0.5$ as in Eq (19).

$$oLRP = \min_s LRP(X, Y_s) \qquad (19)$$

The *moLRP* is the mean optimal localization recall precision error, and *c* denotes the category as in Eq (20).

$$moLRP = \frac{1}{|C|} \sum_{c \in C} oLRP_c \qquad (20)$$

## 4.4 Results analysis

**4.4.1 Learning rate analysis.** To validate the selection of the initial learning rate 1E-5, we conduct parameter experiments comparing five candidate values: {1E-6, 5E-6, 1E-5, 5E-5, 1E-4}, the experimental results are shown in Fig 6. As Fig 6 demonstrates, the initial learning rate 1E-5 strikes an optimal trade-off between convergence speed and stability. The model struggles to converge due to oscillations and the accuracy of small object detection is poor when the learning rate is too large. Conversely, the model training process slows down by an excessively small learning rate, although it does not cause significant performance degradation. These results validate 1E-5 as a robust choice for ELFT to ensure the training process stability.

**4.4.2 Comparative experiments.** To verify the algorithm's effectiveness in improving the detection accuracy of small object images while reducing computational complexity, comparative experiments are conducted on the RSOD, NWPU VHR-10, and PASCAL VOC2007 datasets.

In this paper, the current mainstream object detection algorithms: Faster RCNN, SSD, YOLOv7, Deformable DETR, Conditional DETR, DAB-DETR, DN-DETR, Sparse DETR, and DINO are selected for comparative evaluations on the RSOD dataset. The comparison results are presented in Table 4, where "*" indicates reproduced experimental results, bold indicates the optimal results, and underlined are suboptimal results. As evident from Table 4 that ELFT achieves the mAP of 95.8%, surpassing the baseline algorithm while simultaneously reducing parameters by 10.4% and computational complexity by 22.7%. The moLRP is 0.371, which

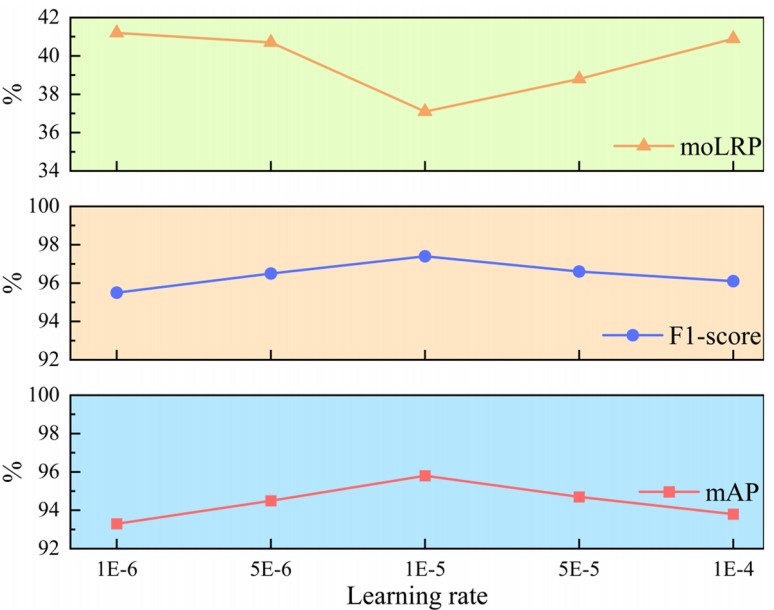

**Fig 6. The effect of different learning rate on ELFT.**

**Table 4. Comparison of detection results of different algorithms in RSOD dataset.** Deformable DETR-D-DETR, Conditional DETR-C-DETR.

| Methods | Params/M | FLOPs/G | F1-score/% | mAP/% | moLRP | moLRP$_{Loc}$ | moLRP$_{FP}$ | moLRP$_{FN}$ |
|---|---|---|---|---|---|---|---|---|
| Faster RCNN [40]* | 28.38 | 941.05 | - | 89.5 | - | - | - | - |
| SSD [41]* | **12.21** | **60.12** | - | 85.5 | - | - | - | - |
| YOLOv7 [42]* | 37.62 | 106.47 | - | 91.6 | - | - | - | - |
| D-DETR [10]* | 38.31 | 203.45 | 96.5 | 94.3 | 0.406 | 0.147 | 0.067 | 0.127 |
| C-DETR [43]* | 40.97 | 98.98 | 95.8 | 93.2 | 0.408 | 0.163 | 0.054 | 0.099 |
| DAB-DETR [11]* | 41.43 | 101.94 | 96.1 | 93.6 | 0.413 | 0.160 | 0.061 | 0.115 |
| DN-DETR [12]* | 41.43 | 101.94 | 96.3 | 93.9 | 0.405 | 0.163 | 0.054 | 0.097 |
| Sparse DETR [18]* | 43.01 | 256.17 | 92.9 | 87.6 | 0.473 | 0.172 | 0.086 | 0.175 |
| DINO [9]* | 45.15 | 289.68 | 96.3 | 93.7 | 0.382 | 0.149 | 0.073 | **0.088** |
| ELFT | 40.44 | 223.88 | **97.4** | **95.8** | **0.371** | **0.146** | **0.032** | 0.100 |

means that the mean optimal localization recall precision error is the smallest, indicating that the bounding box predicted by the ELFT algorithm encircles the target more tightly, and the localization performance is better. These improvements outperform those of the compared representative object detection algorithms. Concretely, the mAP is increased by 2.1% and the F1-score by 1.1% compared to the baseline.

Fig 7 shows the comparison of the Precision-Recall curves for different algorithms on RSOD dataset at an IoU threshold of 0.5. It can be observed that the precision of DINO decreases significantly as recall increases, while ELFT still maintains high precision even at higher recall rates. Meanwhile, ELFT exhibits the largest area under the Precision-Recall curve among all compared mainstream algorithms, demonstrating superior performance. The detection results of different detection algorithms on the RSOD dataset for each category are presented in Table 5. The results reveal that ELFT achieves the highest mAP values in the overpass and playground object categories, while also securing high values in the aircraft and oil tank categories. Compared to various DETR series, ELFT demonstrates different degrees of improvement in mAP. In addition, the mAP surpasses that of representative

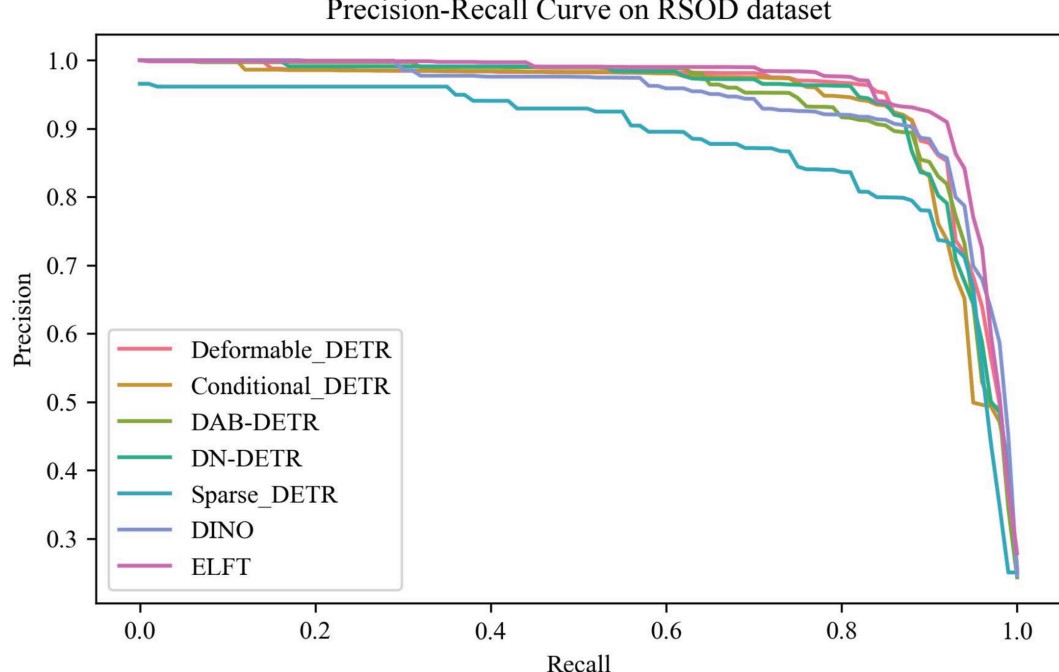

**Fig 7. The precision-recall curve of different algorithms on RSOD dataset.**

**Table 5**. The AP for different detection algorithms on the RSOD dataset.

| Classes | D-DETR | C-DETR | DAB-DETR | DN-DETR | Sparse DETR | Faster R-CNN | SSD | YOLOv7 | DINO | ELFT |
|---|---|---|---|---|---|---|---|---|---|---|
| Aircraft | 93.5 | 90.7 | 92.3 | 92.9 | 81.9 | 70.1 | 56.1 | 90.0 | **95.4** | <u>93.9</u> |
| Oil tank | 96.6 | 97.6 | 97.5 | <u>98.2</u> | 94.7 | 96.9 | 94.4 | **98.5** | 98.0 | 97.4 |
| Overpass | 87.7 | 84.7 | 84.9 | 84.6 | 74.3 | 91.1 | <u>91.6</u> | 78.9 | 81.4 | **91.7** |
| Playground | 99.5 | **100** | 99.7 | <u>99.9</u> | 99.7 | 99.8 | <u>99.9</u> | 99.0 | **100** | **100** |

algorithms of Faster RCNN, SSD, and YOLO series. These results show that ELFT exhibits excellent performance while reducing computational complexity.

Fig 8 illustrates the oLRP, $oLRP_{Loc}$, $oLRP_{FP}$, and $oLRP_{FN}$ of different algorithms on RSOD dataset for each category. It can be seen from the comparison that ELFT achieves the lowest optimal localization recall precision error in the target categories of oil tanks, overpasses, and playgrounds, which proves its effectiveness. The evaluation and comparison results of different algorithms on the NWPU VHR-10 dataset are presented in Table 6. The findings indicate that the proposed algorithm attains an F1-score of 96.1%, which suggests that ELFT strikes a better balance between precision and recall. moLRP, $moLRP_{Loc}$, $moLRP_{FP}$, and $moLRP_{FN}$ are 0.425, 0.168, 0.051, and 0.104, respectively, and mAP reaches 94.3%. It is worth noting that the models outperform the compared mainstream algorithms. More specifically, it improves by 3.3%, 2.9%, 2.4%, 1.8%, 20% and 1.7% in terms of mAP compared to Deformable DETR, Conditional DETR, DAB-DETR, DN-DETR, Sparse DETR and DINO, respectively. The results demonstrate the efficacy and performance of the proposed algorithm. Furthermore, when compared to Faster RCNN, SSD, and YOLOv7, ELFT exhibits a significant increase in mAP by 6.3%, 32.6%, and 4.7%, respectively, indicating its advantages in detection performance.

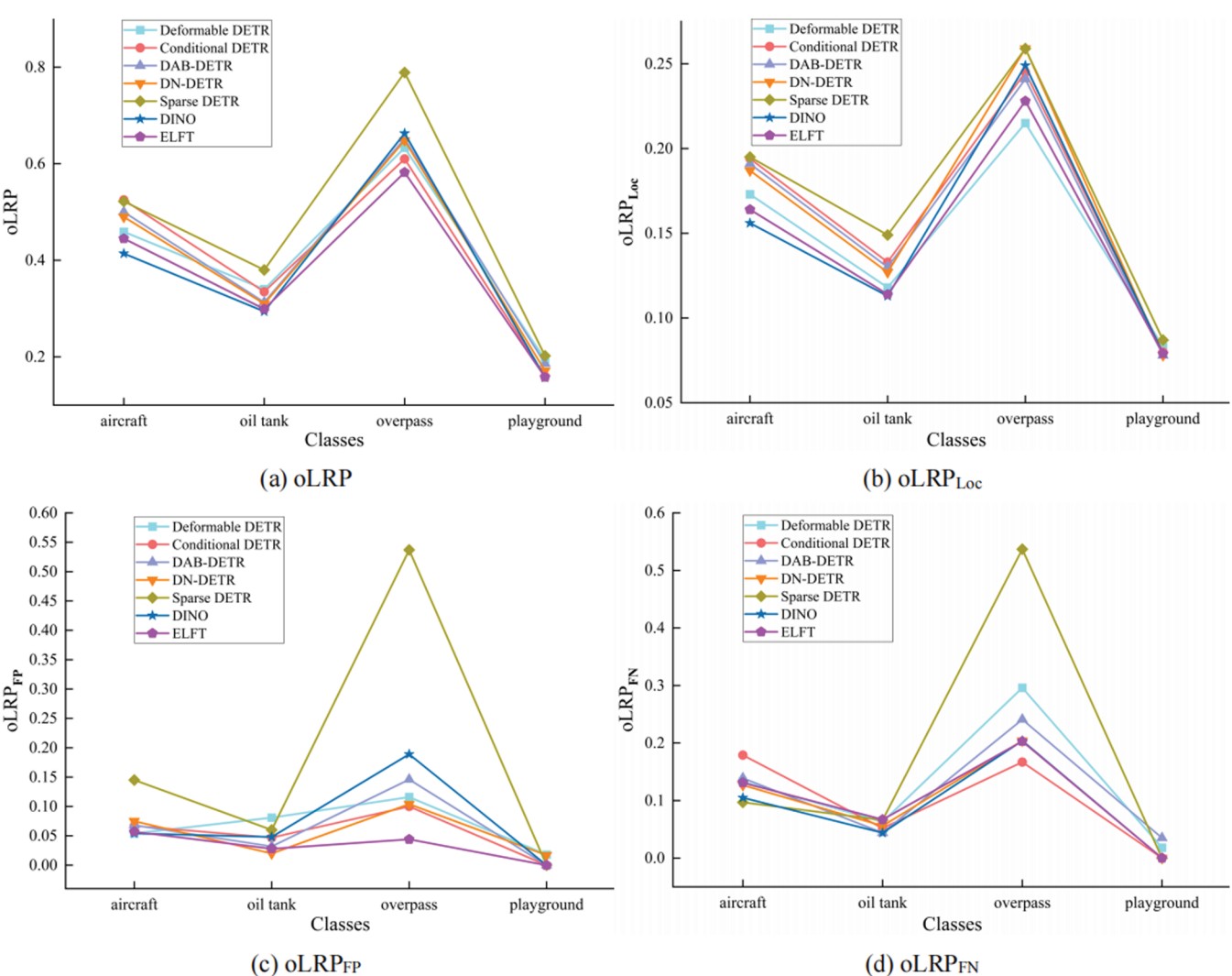

(a) oLRP

(b) oLRP$_{Loc}$

(c) oLRP$_{FP}$

(d) oLRP$_{FN}$

**Fig 8. The optimal localization recall precision (oLRP) of different algorithms on each category of the RSOD dataset.**

**Table 6. Comparison of detection results of different algorithms in NWPU VHR-10 dataset.**

| Methods | Params/M | FLOPs/G | F1-score/% | mAP@0.5/% | moLRP | moLRP$_{Loc}$ | moLRP$_{FP}$ | moLRP$_{FN}$ |
|---|---|---|---|---|---|---|---|---|
| Faster R-CNN [40]* | 28.38 | 941.05 | - | 86.0 | - | - | - | - |
| SSD [41]* | **12.21** | **60.12** | - | 61.7 | - | - | - | - |
| YOLOv7 [42]* | 37.62 | 106.47 | - | 89.6 | - | - | - | - |
| D-DETR [10]* | 38.31 | 203.45 | 93.9 | 91.0 | 0.467 | 0.178 | 0.076 | 0.133 |
| C-DETR [43]* | 40.97 | 98.98 | 93.8 | 91.4 | 0.517 | 0.205 | 0.062 | 0.161 |
| DAB-DETR [11]* | 41.43 | 101.94 | 94.2 | 91.9 | 0.516 | 0.205 | 0.097 | 0.144 |
| DN-DETR [12]* | 41.43 | 101.94 | 94.5 | 92.5 | 0.495 | 0.198 | 0.071 | 0.130 |
| Sparse DETR [18]* | 43.01 | 256.17 | 81.5 | 74.3 | 0.632 | 0.221 | 0.265 | 0.258 |
| DINO [9]* | 45.15 | 289.68 | 94.9 | 92.6 | 0.454 | 0.181 | **0.050** | 0.125 |
| ELFT | 40.44 | 223.88 | **96.1** | **94.3** | **0.425** | **0.168** | 0.051 | **0.104** |

Fig 9 presents the Precision-Recall curves of different algorithms on the NWPU VHR-10 dataset. ELFT significantly outperforms other algorithms in the high recall region and reduces the missed detection. These results indicate that ELFT effectively balances the accuracy and recall, and is suitable for small object detection scenarios. The detection results of different detection algorithms on the NWPU VHR-10 dataset for each category are reported in Table 7. Through the analysis of Table 7, it is apparent that the ELFT has achieved the highest value in detecting four types of small objects, such as aircraft, basketball courts, harbors, and vehicles. Additionally, it achieves suboptimal values in detecting three types of small objects: oil tanks, tennis courts, and bridges. The possible reason for the highest or suboptimal performance in ships and ground track fields lies in that the proportion of these objects in the images is tiny, and the resolution may be low. Regarding baseball diamonds, their features are similar to those of the surrounding environment, making it challenging to extract distinct features, which is prone to incur lower detection accuracy.

Fig 10 illustrates the performance of different algorithms for oLRP, $oLRP_{Loc}$, $oLRP_{FP}$ and $oLRP_{FN}$ on each category in the NWPU VHR-10 dataset. It shows that the optimal localization recall precision error is the smallest in all categories except the storage tank target category, suggesting that ELFT predicts a higher degree of tightness of the bounding box enclosing the target. The proposed algorithm and DINO are compared to some current representative object detection algorithms in the PASCAL VOC2007 dataset, and the experimental results are displayed in Table 8. The findings show that the proposed algorithm achieves an F1-score of 91.3% and the mAP of 85.3%, outperforming other algorithms in both metrics. ELFT achieves a lower optimal localization recall precision error (oLRP) of only 0.475 compared to other algorithms, thereby proving its effectiveness. Specifically, the F1-score is increased by 0.5% at least and 5.3% at most compared to other algorithms, while the mAP is

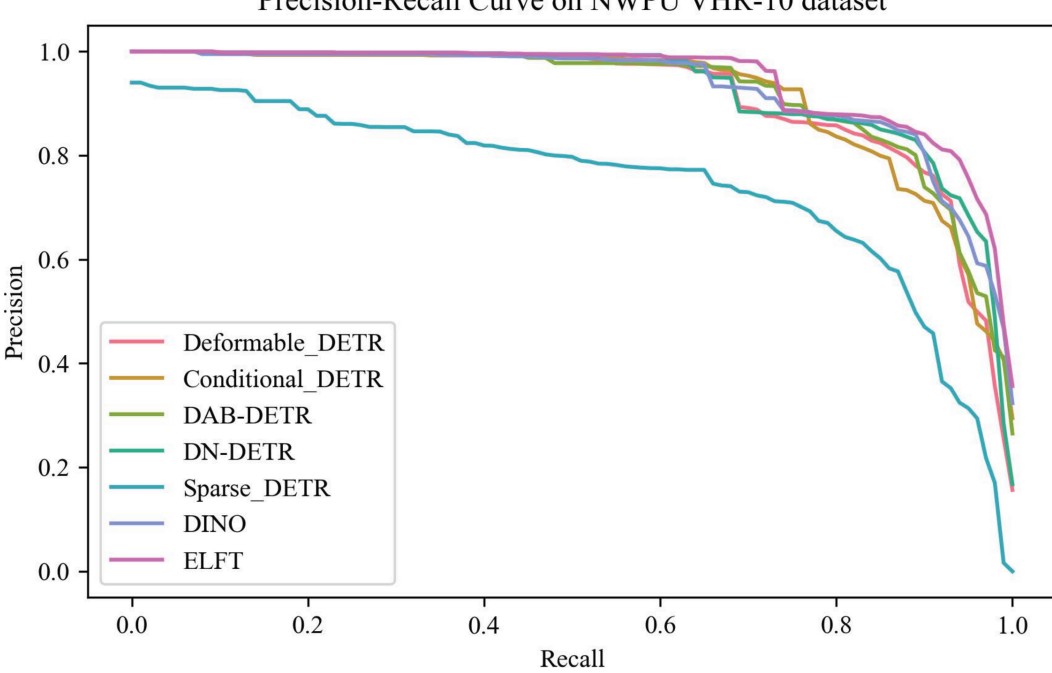

**Fig 9. The precision-recall curve of different algorithms on NWPU VHR-10 dataset.**

**Table 7. The AP for different detection algorithms on the NWPU VHR-10 dataset.** Airplane-APL, Ship-SH, Storage tank-ST, Baseball diamond-BD, Tennis court-TC, Basketball court-BC, Ground track field-GTF, Harbor-HA, Bridge-BR, Vehicle-VE.

| Classes | D-DETR | C-DETR | DAB-DETR | DN-DETR | Sparse DETR | Faster RCNN | SSD | YOLOv7 | DINO | ELFT |
|---|---|---|---|---|---|---|---|---|---|---|
| APL | 98.6 | 98.8 | **98.9** | 98.6 | 88.3 | 97.9 | 87.3 | 98.2 | **98.9** | **98.9** |
| SH | 89.7 | 90.2 | 92.5 | **96.4** | 84.9 | 76.4 | 52.8 | 85.0 | 89.6 | 91.7 |
| ST | 93.4 | 91.4 | 89.2 | 93.8 | 88.5 | 77.1 | 11.4 | 92.2 | **95.7** | 95.5 |
| BD | 96.9 | 99.5 | **99.6** | 97.5 | 95.7 | 99.2 | 94.8 | 97.2 | 99.1 | 98.9 |
| TC | 96.5 | 94.3 | 96.8 | 96.7 | 77.7 | 92.7 | 60.5 | 97.4 | **97.9** | 97.7 |
| BC | 90.3 | 97.7 | 96.8 | 95.6 | 59.9 | 91.9 | 58.6 | 80.5 | 97.4 | **99.3** |
| GTF | 98.5 | 99.6 | 99.7 | **99.8** | 90.1 | 99.6 | 99.7 | 98.8 | **99.8** | 99.2 |
| HA | 93.4 | 88.7 | 93.8 | 94.9 | 67.8 | 94.3 | 73.2 | 94.3 | 94.7 | **96.8** |
| BR | 67.8 | **74.2** | 67.9 | 66.9 | 10.9 | 72.6 | 63.8 | 66.5 | 65.0 | 73.1 |
| VE | 85.2 | 79.9 | 84.3 | 84.8 | 79.0 | 58.1 | 14.8 | 85.5 | 88.0 | **92.2** |

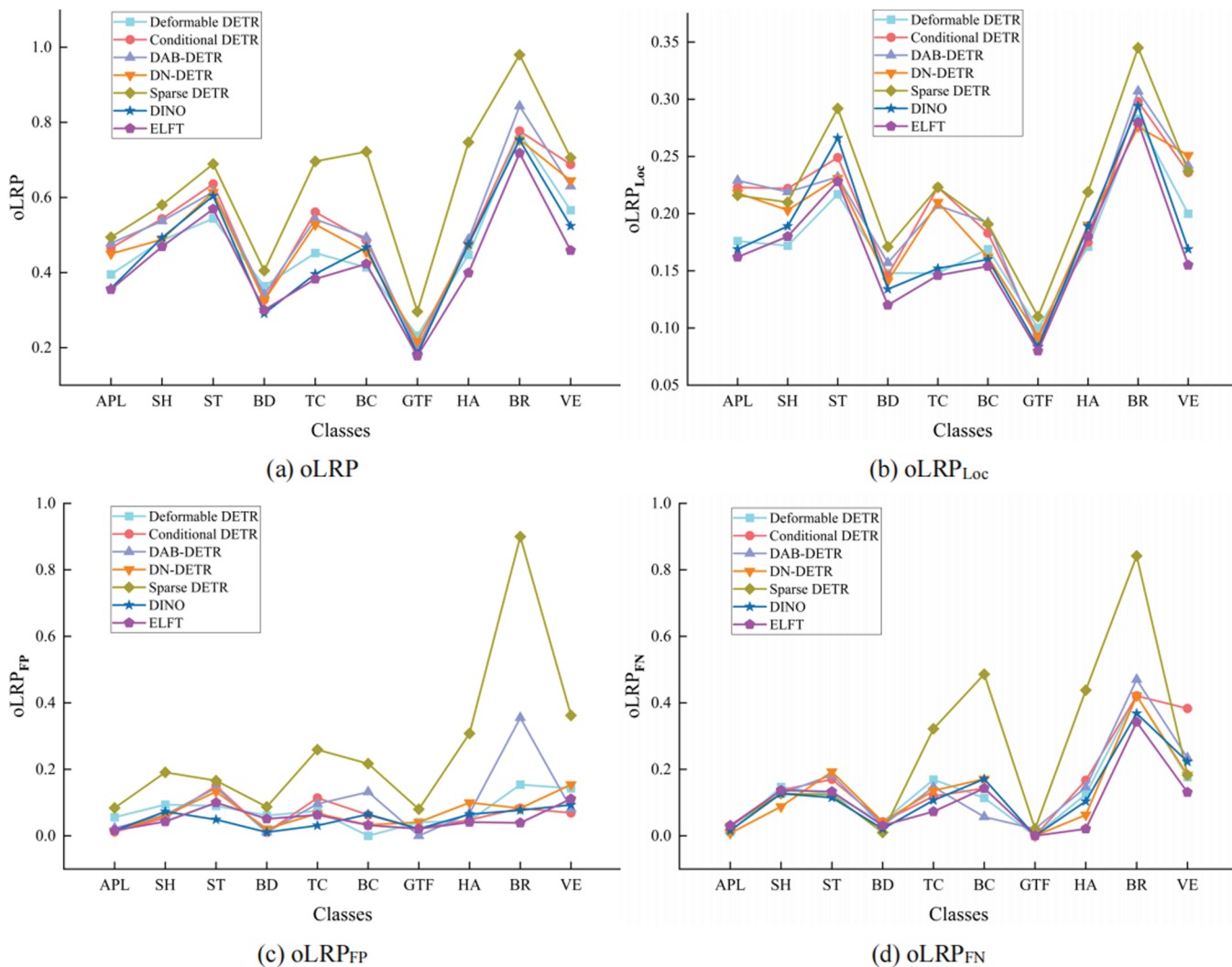

**Fig 10. The optimal localization recall precision (oLRP) of different algorithms on each category of the NWPU VHR-10 dataset.**

**Table 8**. Contrast of detection results of different algorithms in PASCAL VOC2007 dataset.

| Methods | Params/M | FLOPs/G | F1-score/% | mAP@0.5/% | moLRP | moLRP$_{Loc}$ | moLRP$_{FP}$ | moLRP$_{FN}$ |
|---|---|---|---|---|---|---|---|---|
| Faster RCNN [40] | - | - | - | 69.9 | - | - | - | - |
| SSD512 [41] | - | - | - | 71.6 | - | - | - | - |
| YOLOX [44]* | **5.04** | **15.26** | - | 77.0 | - | - | - | - |
| D-DETR [10]* | <u>38.31</u> | 203.45 | 88.6 | 82.3 | 0.517 | 0.126 | 0.155 | 0.280 |
| C-DETR [43]* | 40.97 | <u>98.98</u> | 86.0 | 81.9 | 0.528 | 0.136 | 0.147 | 0.287 |
| DAB-DETR [11]* | 41.43 | 101.94 | 89.8 | 83.3 | 0.518 | 0.131 | 0.153 | 0.274 |
| DN-DETR [12]* | 41.43 | 101.94 | 90.6 | 84.4 | 0.500 | 0.123 | 0.143 | 0.268 |
| Sparse DETR [18]* | 43.01 | 256.17 | 86.9 | 79.7 | 0.534 | 0.130 | 0.171 | 0.294 |
| AFCNet [45] | - | - | - | 81.9 | - | - | - | - |
| DINO [9]* | 45.15 | 289.68 | <u>90.8</u> | <u>84.6</u> | <u>0.483</u> | <u>0.118</u> | <u>0.131</u> | **0.260** |
| ELFT | 40.44 | 223.88 | **91.3** | **85.3** | **0.475** | **0.113** | **0.120** | <u>0.265</u> |

increased by 0.7% at least and up to 15.4%. The moLRP is reduced by a minimum of 0.008 and a maximum of 0.059 compared to other algorithms, further demonstrating the high performance of the ELFT algorithm in localization, regression and classification.

Fig 11 shows the Precision-Recall curves of different algorithms on the PASCAL VOC2007 dataset. From the curve patterns, the Precision-Recall curves of all the compared algorithms on the PASCAL VOC2007 dataset exhibit a gently decreasing trend. However, the Precision-Recall curve of ELFT always ranks above the other algorithms, which indicates that ELFT maintains a higher accuracy across the entire recall rate range. The detection results for different classes in the PASCAL VOC2007 dataset on different detection algorithms are presented in Table 9. The results reveal that among the 20 categories in the VOC dataset, except for the

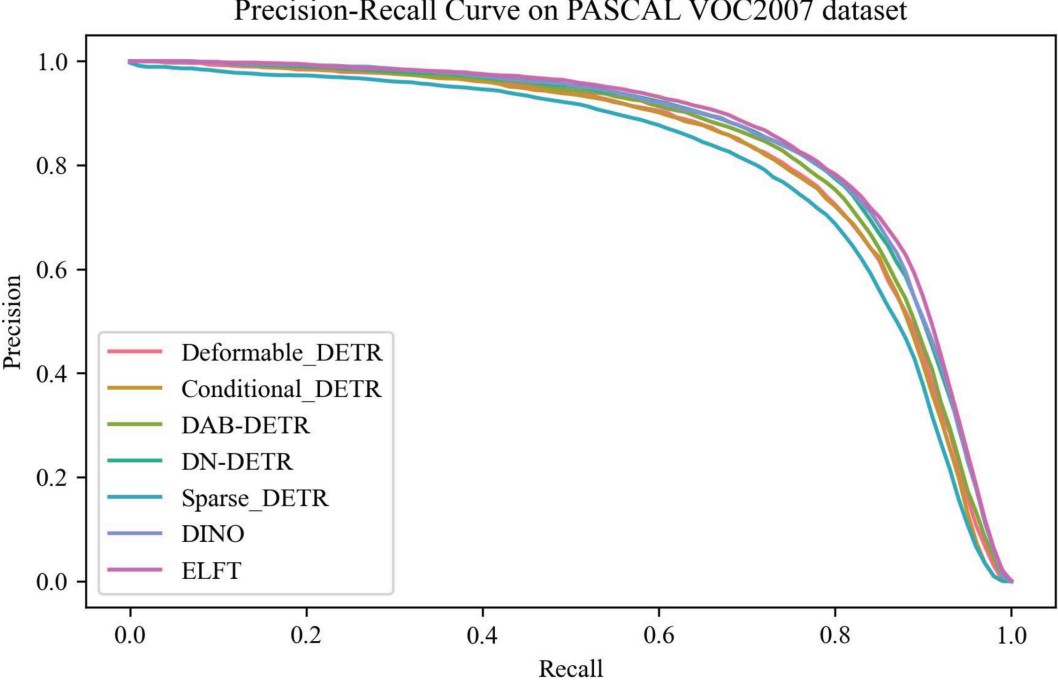

**Fig 11. The precision-recall curve of different algorithms on PASCAL VOC2007 dataset.**

**Table 9. The AP for different detection algorithms on the PASCAL VOC2007 dataset.**

| Classes | D-DETR | C-DETR | DAB-DETR | DN-DETR | Sparse DETR | Faster RCNN | SSD512 | YOLOX | DINO | ELFT |
|---|---|---|---|---|---|---|---|---|---|---|
| aero | 88.3 | 90.6 | 91.3 | **93.3** | 88.9 | 70.0 | 75.1 | 83.2 | 91.8 | <u>92.4</u> |
| bike | 86.1 | 84.7 | 86.4 | 87.6 | 86.3 | 80.6 | 81.4 | 85.4 | <u>88.0</u> | **88.5** |
| bird | 82.1 | 84.1 | 82.7 | 84.4 | 83.0 | 70.1 | 69.8 | 72.9 | **86.7** | <u>86.3</u> |
| boat | 73.8 | 72.9 | 74.3 | 75.4 | 68.7 | 57.3 | 60.8 | 66.0 | <u>77.4</u> | **78.5** |
| bottle | 72.3 | 65.6 | 73.5 | 72.1 | 67.4 | 49.9 | 46.3 | 66.0 | <u>75.3</u> | **77.0** |
| bus | <u>87.3</u> | 86.2 | 86.4 | 86.1 | 85.7 | 78.2 | 82.6 | 83.2 | 86.8 | **88.6** |
| car | 90.9 | 88.7 | 90.2 | <u>91.3</u> | 89.3 | 80.4 | 84.7 | 89.3 | <u>91.3</u> | **91.7** |
| cat | 88.6 | 91.2 | <u>92.8</u> | 92.7 | 89.1 | 82.0 | 84.1 | 79.2 | 92.1 | **94.0** |
| chair | 67.7 | 66.0 | 68.2 | **70.9** | 62.9 | 52.2 | 48.5 | 62.2 | 69.0 | <u>70.0</u> |
| cow | 88.9 | 88.2 | 88.2 | 91.3 | 89.8 | 75.3 | 75.0 | 80.4 | **91.7** | <u>91.6</u> |
| table | 70.0 | 72.1 | 74.6 | **75.7** | 60.6 | 67.2 | 67.4 | <u>75.3</u> | 73.3 | 74.1 |
| dog | 90.4 | 89.3 | 90.3 | 90.0 | 86.0 | 80.3 | 82.3 | 79.1 | <u>91.9</u> | **92.3** |
| horse | 91.5 | 90.8 | 90.9 | **93.6** | 90.1 | 79.8 | 83.9 | 87.0 | <u>92.8</u> | 92.1 |
| motorbike | 88.1 | 85.7 | 89.6 | <u>90.8</u> | 85.6 | 75.0 | 79.4 | 84.3 | 90.1 | **91.2** |
| person | 89.9 | 88.3 | 89.4 | <u>90.2</u> | 88.9 | 76.3 | 76.6 | 86.0 | 90.0 | **90.6** |
| plant | 60.2 | 61.8 | 62.8 | **66.8** | 58.4 | 39.1 | 44.9 | 53.4 | 63.8 | <u>64.3</u> |
| sheep | 79.7 | 82.0 | 83.5 | 83.6 | 79.8 | 68.3 | 69.9 | 75.3 | <u>85.9</u> | **88.2** |
| sofa | 75.1 | 77.4 | 77.0 | **79.0** | 63.2 | 67.3 | 69.1 | 71.4 | 77.8 | <u>78.3</u> |
| train | 90.7 | 89.4 | 88.9 | 89.9 | 85.6 | 81.1 | 78.1 | 81.7 | <u>90.8</u> | **90.9** |
| tv | 83.4 | 83.8 | 85.0 | 84.0 | 85.7 | 67.6 | 71.8 | 78.3 | **86.5** | <u>86.1</u> |

categories of table and horse, the remaining 18 categories have achieved optimal or suboptimal detection results compared to the other algorithms. One possible reason is that the number of positive sample images for tables and horses in the dataset is relatively few, which limits the ability of the algorithm to learn and recognize features for these two categories.

Fig 12 shows the performance of different algorithms for oLRP, $oLRP_{Loc}$, $oLRP_{FP}$, and $oLRP_{FN}$ in each category of the PASCAL VOC2007 dataset. It can be seen that ELFT exhibits less error in the optimal localization recall precision, which proves the superiority of ELFT's performance.

**4.4.3 Ablation experiments.** To validate the efficiency of different components, this paper performs ablation experiments on the RSOD dataset, comparing the performance of algorithms in different cases. Experiment A represents the DINO algorithm, experiments B to F represent the addition of different improved methods. "√" indicates the use of a method, while "×" is not used. The experimental results are summarized in Table 10.

The core idea of the GFUM module is that the six-layer encoder is divided into three groups. For each group, the first layer aggregates information globally through high-level features to reduce computational load, while the second layer preserves details using low-level features, achieving a balance between multi-scale feature maintenance and computational efficiency. Moreover, the ELGFA moudle by pooling image features from the height, width, and global dimensions, effectively integrates both local and global contextual information to enhance the input feature image, aiming to accurately identify regions of interest. Furthermore, the CB module computes the average of feature vectors across layers and then rebroadcasts this global context information to each feature vector. Comparing experiment A and experiment B, we observe that adding GFUM to the encoder in DINO reduces FLOPs by 21.0% with a slight increase in mAP, demonstrating that GFUM effectively decreases the computational load of the encoder by alternately updating high and low-level features. When comparing experiment B with experiment C, we observe that a reduction of 10.4% has been achieved in the total number of parameters within the algorithm, with a further reduction

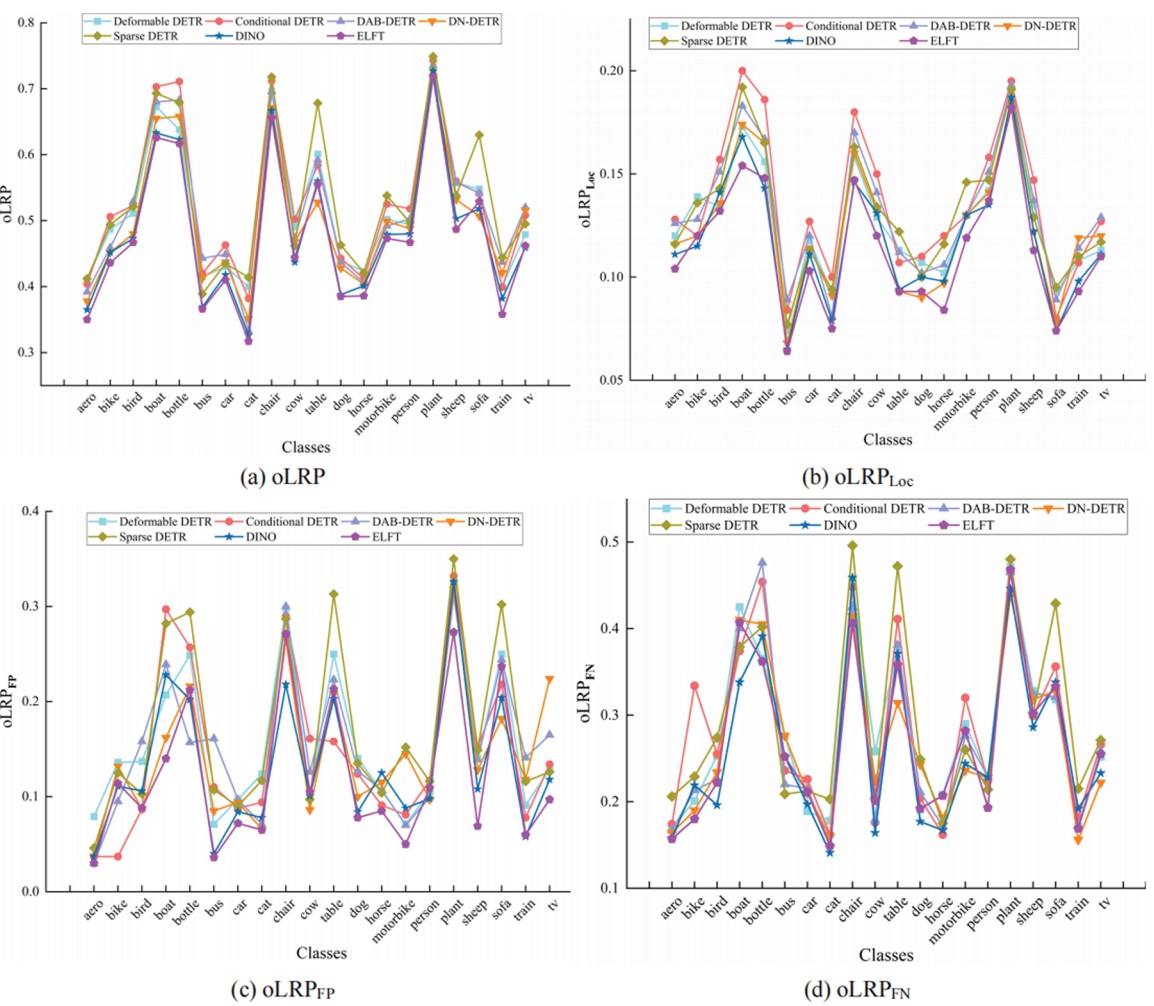

**Fig 12. The optimal localization recall precision (oLRP) of different algorithms on each category of the PASCAL VOC2007 dataset.**

**Table 10. The results of ablation studies on the RSOD dataset.**

| Groups | baseline | GFUM | ELA | ELGFA | CB | mAP/% | Params/M | FLOPs/G |
|--------|----------|------|-----|-------|----|-------|----------|---------|
| A | √ | × | × | × | × | 93.7 | 45.15 | 289.68 |
| B | √ | √ | × | × | × | 93.8 | 45.15 | 228.84 |
| C | √ | √ | √ | × | × | 93.9 | 40.44 | 223.88 |
| D | √ | √ | × | √ | × | 94.5 | 40.44 | 223.88 |
| E | √ | √ | × | × | √ | 94.7 | 45.15 | 228.84 |
| F | √ | √ | × | √ | √ | 95.8 | 40.44 | 223.88 |

in FLOPs to 223.88G. Notably, the mAP is also slightly improved. The ELA module precisely identifies the region's position of interest, preserves the input feature channels' dimensionality, and remains lightweight. Furthermore, the comparison between experiment B and experiment D reveals that the ELGFA module considers the global context information more comprehensively, improves the missed detection issue of small objects, and increases the mAP to 94.5%. The comparison between experiment B and experiment E shows that the CB module further boosts the mAP by 0.9%, thereby proving the effectiveness of the CB module in

the algorithm. Experiment F combines three improved methods, and when compared to the original DINO algorithm, a reduction of 10.4% has been achieved in the total number of parameters, the FLOPs are reduced by 22.7%, and the mAP is increased by 2.1%. Note that the experiment F performs the best in regards of mAP, parameters and FLOPs as well. Therefore, experimental results confirm the effectiveness of ELFT.

To evaluate the versatility of the ELGFA module, we replace the backbone of the benchmark model DINO with ResNet101. By incorporating the ELGFA module, we conduct experiments on three datasets, and the experimental results are shown in Table 11. It can be seen that when ResNet50 is used as the backbone, the FLOPs are 289.68G, parameters are 45.15M, and the mAP in the three datasets is 93.7%, 92.6%, and 84.6%, respectively. After adding the ELGFA module, the FLOPs are reduced to 284.80G, the Params are decreased to 40.44M, and the mAP in the three datasets is increased to 95.4%, 93.5%, and 84.8%, respectively. For ResNet101 backbone configurations, the baseline yields 369.23G FLOPs, 64.09M parameters, and mAP is 95.6%, 93.3%, and 84.7%, respectively. After adding the ELGFA module, FLOPs are reduced to 364.28G, parameters are decreased to 59.38M, and mAP on the three datasets is boosted to 96.1%, 93.7%, and 85.0%, respectively. These results demonstrate that the ELGFA module is applicable in different models and can effectively improve the small object detection accuracy. In addition, the experimental results show that ResNet101 outperforms ResNet50 marginally, but it requires higher computational cost. This is the reason why we choose ResNet50 as the backbone network.

Furthermore, to validate the additional detector generality of the ELGFA module, we integrated it into three state-of-the-art DETR variants, i.e. Deformable DETR, DAB-DETR, and DN-DETR. The results of the experiments conducted on the RSOD dataset are summarized in Table 12. It is clear to see that ELGFA integration consistently improves small object detection performance across all evaluated frameworks while reducing computational overhead. Specifically, embedding ELGFA module into Deformable DETR reduces parameter count by 4.71M and FLOPs to 198.50G, F1-score and mAP are improved by 0.2%, 0.6%, respectively. Similarly, DAB-DETR with ELGFA achieves an 11.4% parameter reduction and reduces FLOPs to 96.98G, F1-score and mAP gain of 1.0%, 1.9%, respectively. DN-DETR with ELGFA decreases parameter count to 36.72M, reduces FLOPs by 4.9%, and improves F1-score and mAP by 0.9%, 1.7%, respectively. The mean optimal localization recall precision (moLRP) is decreased to 0.395, 0.392, and 0.383, respectively. These findings demonstrate the generalizability of the ELGFA module, effectively improving small object detection performance while maintaining computational efficiency across diverse detectors.

## 4.5 Visualizations

To thoroughly investigate ELFT's performance in small object detection, we perform Grad-CAM visualization for heatmap analysis, as shown in Fig 13. It intuitively shows how the

**Table 11. Comparison of different models with the ELGFA module.**

| backbone | Params/M | FLOPs/G | mAP/% | | |
|---|---|---|---|---|---|
| | | | RSOD | NWPU VHR-10 | PASCAL VOC2007 |
| ResNet50 | 45.15 | 289.68 | 93.7 | 92.6 | 84.6 |
| ResNet50+ELGFA | 40.44 | 284.80 | 95.4 | 93.5 | 84.8 |
| ResNet101 | 64.09 | 369.23 | 95.6 | 93.3 | 84.7 |
| ResNet101+ELGFA | 59.38 | 364.28 | 96.1 | 93.7 | 85.0 |

**Table 12. The performance of the ELGFA module in different detectors.**

| Detectors | Params/M | FLOPs/G | F1-score/% | mAP/% | moLRP |
|---|---|---|---|---|---|
| Deformable DETR | 38.31 | 203.45 | 96.5 | 94.3 | 0.406 |
| Deformable DETR+ELGFA | 33.60 | 198.50 | 96.7 | 94.9 | 0.395 |
| DAB-DETR | 41.43 | 101.94 | 96.1 | 93.6 | 0.413 |
| DAB-DETR+ELGFA | 36.72 | 96.98 | 97.1 | 95.5 | 0.392 |
| DN-DETR | 41.43 | 101.94 | 96.3 | 93.9 | 0.405 |
| DN-DETR +ELGFA | 36.72 | 96.98 | 97.2 | 95.6 | 0.383 |

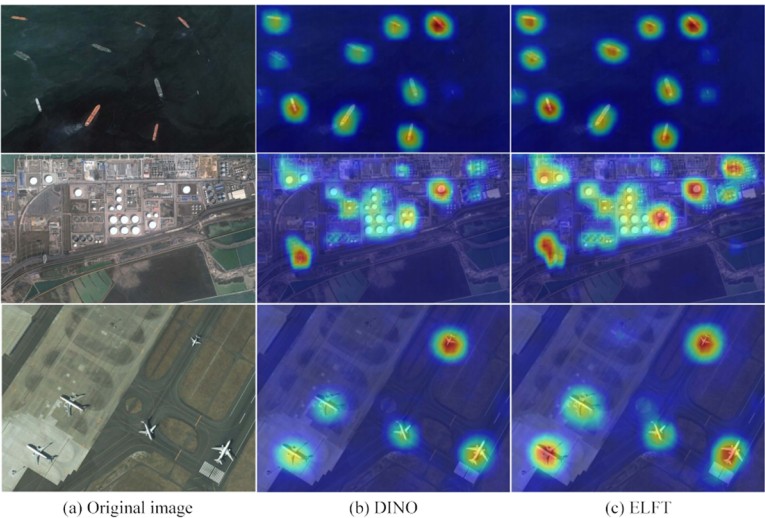

(a) Original image (b) DINO (c) ELFT

**Fig 13. Grad-CAM visualization of the learned features.**

model localizes specific regions. The Grad-CAM technique enhances visualization of the model's decision-making process, which aids in understanding how targets are identified within images. For small object detection, ELFT demonstrates distinct strengths. Experimental results indicate it successfully detects small objects missed by the original algorithm. This further confirms that the improved method has higher sensitivity to small objects, contributing to enhanced detection accuracy. The detection effects of DAB-DETR, DINO, and the proposed algorithm are shown in Fig 14, where wrongly detected or missed objects are highlighted with a red circle. As Fig 14(a) illustrates, DAB-DETR falsely detects two boarding bridges as aircraft, and the original DINO similarly falsely detects the boarding bridges at the top right of the picture as aircraft. In contrast, ELFT accurately detects all objects within the image without any false detections. As Fig 14(b) shows, the four tennis courts in the upper right corner are not detected by DAB-DETR, and the lower tennis courts are not detected by DINO. On the contrary, ELFT successfully detects all objects. Due to the compact distribution of tennis courts and the similar colors and backgrounds, it is difficult to detect. The CB module embedded in the proposed algorithm considers the global context of the entire image, thereby significantly enhancing the detection performance in dense environments. In Fig 14(c), both DAB-DETR and the original DINO algorithm fail to detect the four oil tank objects located at the top right. Conversely, ELFT accurately detects these objects, indicating that the ELGFA module proposed in this paper can better extract the location features of objects in the image. This fusion of feature map details effectively solves the issue

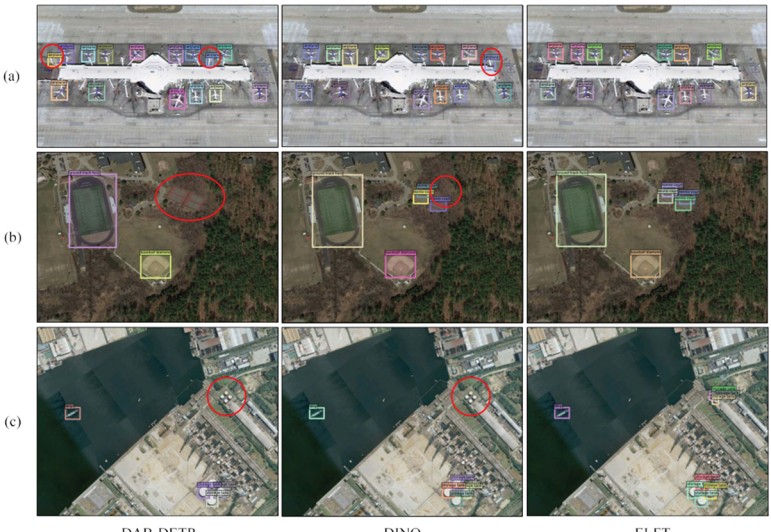

**Fig 14. Comparison of detection results on NWPU VHR-10 dataset.**

of missed detections of oil tank objects. Figs 15 and 16 present the detection results on the RSOD dataset and PASCAL VOC2007 dataset, respectively.

## 4.6 Discussion

Although the proposed model demonstrates competitive performance in both quantitative metrics and qualitative visualization, somre failure cases of our method appear in several scenarios, e.g., low-light scenes or severe object occlusion. For low-light, the reduction in image contrast and color fidelity limits the discriminative capacity of visual features, leading to suboptimal predictions. For severe object occlusion, partial visibility of key semantic regions reduces the effectiveness of local feature extraction, making the representation less reliable. We attribute these degradations to two main factors: First, Data distribution bias, as the training dataset contains fewer samples representing extreme lighting or heavy occlusion conditions. Second, Limitations in spatial dependency modeling, where the current mechanism may not fully capture long-range relationships or adapt to highly variable object appearances. At present, transfer leanring and progressive hierarchical attention mechanisms has applied to address these problems. It is an interesting direction to apply these paradigms into small object detection in the future.

## 5 Conclusion

In this paper, we present an ELFT object detection algorithm, which obtains a balance between the computational cost and detection performance. The ResNet50 structure is augmented with the ELGFA module to efficiently capture long-range spatial relationships and integrate global and local contextual data. The GFUM is devised to optimize the efficiency of the encoder. Furthermore, we adopt the CB module to obtain more extensive and detailed context information, which makes the localization and recognition of small objects more accurate. Extensive experiments validate the efficacy of the proposed approach in enhancing the accuracy of detecting small objects and reducing parameter count and computational cost.

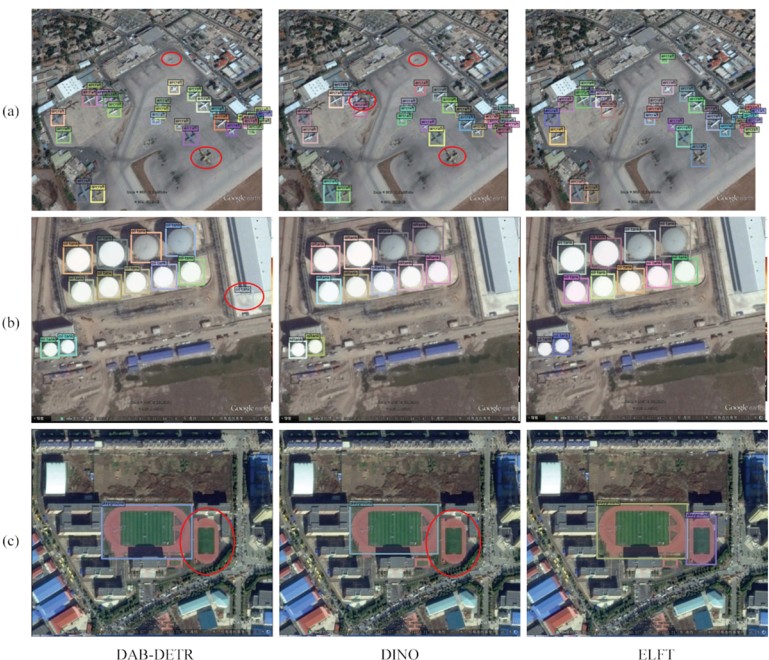

DAB-DETR　　　　　DINO　　　　　ELFT

**Fig 15. Comparison of detection results on RSOD dataset.**

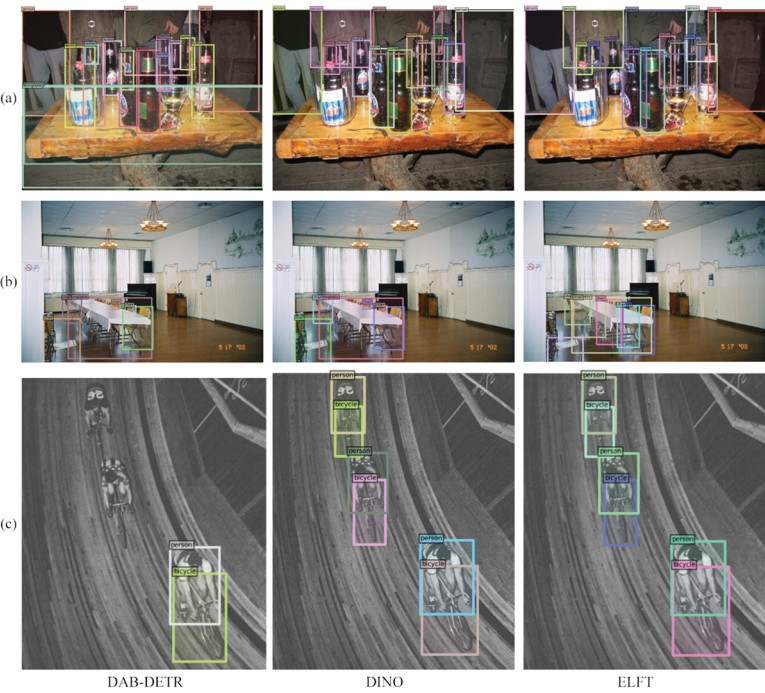

DAB-DETR　　　　　DINO　　　　　ELFT

**Fig 16. Comparison of detection results on PASCAL VOC2007 dataset.**

Since the algorithm mainly relies on a Transformer structure, there is ample margin to simplify it due to the quadratic computational complexity of self-attention. We will make further efforts to explore the lightweight methods and significantly improve the detection ability for small objects in future work.

## Author contributions

**Conceptualization:** Guoguang Hua, Li Li.

**Data curation:** Fangfang Wu, Li Li.

**Formal analysis:** Guoguang Hua, Fangfang Wu, Li Li.

**Funding acquisition:** Li Li.

**Investigation:** Li Li.

**Methodology:** Guoguang Hua, Fangfang Wu, Li Li.

**Project administration:** Guoguang Hua, Li Li.

**Resources:** Li Li.

**Software:** Guoguang Hua, Fangfang Wu, Guangzhao Hao, Chenbo Xia, Li Li.

**Supervision:** Li Li.

**Validation:** Guoguang Hua, Fangfang Wu, Guangzhao Hao, Chenbo Xia, Li Li.

**Visualization:** Fangfang Wu, Chenbo Xia.

**Writing – original draft:** Fangfang Wu.

**Writing – review & editing:** Guoguang Hua, Fangfang Wu, Li Li.

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
