## [Decision Letter · Decision Letter 0]

2 Jul 2025

PONE-D-25-27252ELFT: efficient local-global fusion Transformer for small object detectionPLOS ONE

Dear Dr. Li,

Thank you for submitting your manuscript to PLOS ONE. After careful consideration, we feel that it has merit but does not fully meet PLOS ONE’s publication criteria as it currently stands. Therefore, we invite you to submit a revised version of the manuscript that addresses the points raised during the review process.

We look forward to receiving your revised manuscript.

Kind regards,

Aiqing Fang

Academic Editor

PLOS ONE

 [The work was funded by Science and Technology Research and Development Plan Project of Handan, Hebei Province, China (21422031289) and Science Research Project of Hebei Education Department, China (SQ2023096).]. 

Additional Editor Comments (if provided):

Reviewers' comments:

Reviewer's Responses to Questions

**Comments to the Author**

1. Is the manuscript technically sound, and do the data support the conclusions?

Reviewer #1: Yes

Reviewer #2: Yes

Reviewer #3: Yes

2. Has the statistical analysis been performed appropriately and rigorously? 

Reviewer #1: Yes

Reviewer #2: Yes

Reviewer #3: Yes

3. Have the authors made all data underlying the findings in their manuscript fully available?

Reviewer #1: Yes

Reviewer #2: Yes

Reviewer #3: Yes

4. Is the manuscript presented in an intelligible fashion and written in standard English?

Reviewer #1: Yes

Reviewer #2: Yes

Reviewer #3: Yes

5. Review Comments to the Author

Reviewer #1: Abstract:The abstract should begin by clearly stating the specific challenges in small object detection. It is recommended to briefly summarize the main difficulties, such as insufficient semantic information, occlusion, and low pixel ratio for small objects. Furthermore, the advantages and limitations of conventional approaches should be discussed in relation to these challenges, emphasizing why further improvements are needed. The authors should clarify the concrete improvements proposed in this work that specifically address these shortcomings.

Introduction: In the introduction, while the authors discuss improvements in global feature extraction, it is suggested that they compare global and local feature extraction methods, particularly within deep learning frameworks. For instance, works such as "CSRM-MIM: A Self-Supervised Pre-training Method for Detecting Catenary Support Components in Electrified Railways" and "Surrogate modeling of pantograph-catenary system interactions" should be referenced to highlight both global and local strategies. Additionally, it would be valuable to compare with self-supervised feature extraction approaches, including but not limited to "CSRM-MIM: A Self-Supervised Pre-training Method for Detecting Catenary Support Components in Electrified Railways" and "Multi-modal imitation learning for arc detection in complex railway environments".

Section 3.1 Network Architecture:The manuscript should provide a clear description of the inputs and outputs between modules. For Figure 1, it is suggested to use a network architecture diagram of DINO rather than a process flowchart for better clarity. Given that the code is not open-sourced, it is also advisable for the authors to provide a pseudo-algorithm to facilitate understanding and reproducibility.

Similarly, Figure 2 should indicate the input and output variables between the modules to enhance the comprehensibility of the network's structure.

In the section describing the "Efficient Local-Global Fusion Transformer Algorithm", the manuscript should provide explicit details of the model's loss function and optimization process.

Experimental Section : Besides quantitative evidence, the authors are encouraged to provide qualitative evaluations to demonstrate the superiority of the proposed model, such as attention or heat maps to visualize the learned features and support the claims.

It is recommended to add a discussion subsection in the experimental part, analyzing scenarios where the model's performance degrades. The authors should discuss possible reasons for such degradation and outline potential directions for future work to address these limitations.

Overall, this is an interesting and valuable study. However, substantial revisions are required before publication.

Reviewer #2: In the Abstract section of the manuscript, the sentence "Transformer-based algorithms have exhibited remarkable performance" could be improved for clarity and specificity. A refined version of this sentence would be: "Transformer-based algorithms have demonstrated remarkable performance in the domain of computer vision tasks." This revision explicitly mentions the domain where the performance is notable, providing a clearer context for the reader and emphasizing the relevance of the research in computer vision. Additionally, please note that we have reviewed your submission and require you to remove all figures from within your manuscript file, leaving only the individual TIFF/EPS image files. These files will be automatically included in the reviewer’s PDF. This process ensures that the images are of high quality and properly formatted for the review process.

Reviewer #3: The authors have made commendable progress in designing a lightweight model tailored for effective detection of small objects.

The manuscript is generally well-structured and clearly presented. The visualizations, especially the detection outputs, are intuitive and support the claims made.

However, the ablation study section would benefit from a more in-depth technical explanation. Clarifying the rationale behind each component tested, along with a more rigorous discussion of performance variations, would enhance the overall scientific contribution.

The results are presented effectively. Nevertheless, for completeness and consistency, it is recommended that detection outcomes across all datasets be visualized similarly to Figure 13.

6. PLOS authors have the option to publish the peer review history of their article (what does this mean?). If published, this will include your full peer review and any attached files.

Reviewer #1: No

Reviewer #2: No

Reviewer #3: No

---

## [Author Response · Author response to Decision Letter 1]

19 Aug 2025

Reviewer #1

1.Introduction: In the introduction, while the authors discuss improvements in global feature extraction, it is suggested that they compare global and local feature extraction methods, particularly within deep learning frameworks. For instance, works such as "CSRM-MIM: A Self-Supervised Pre-training Method for Detecting Catenary Support Components in Electrified Railways" and "Surrogate modeling of pantograph-catenary system interactions" should be referenced to highlight both global and local strategies. Additionally, it would be valuable to compare with self-supervised feature extraction approaches, including but not limited to "CSRM-MIM: A Self-Supervised Pre-training Method for Detecting Catenary Support Components in Electrified Railways" and "Multi-modal imitation learning for arc detection in complex railway environments".

Answer We thank the reviewer for this valuable suggestion regarding the comparison between global and local feature extraction methods. Firstly, We have added the works "CSRM-MIM: A Self-Supervised Pre-training Method for Detecting Catenary Support Components in Electrified Railways" and "Surrogate modeling of pantograph-catenary system interactions" to illustrate representative global and local strategies in related domains. Besides, We have also referenced "Multi-modal imitation learning for arc detection in complex railway environments" and other relevant works to highlight recent trends in self-supervised learning for feature extraction, positioning our work within this broader research landscape. In the revised manuscript, we have expanded the Introduction to:

Some approaches design networks via the integration of local and global strategies, aimed at extracting both global and local features. For example, the Fourier Neural Operator with Local Priors and Global Perceptron (FOLGP) proposed in [2] integrates Transformer modules to model global contextual information across frequency bands, enhancing the capture of correlations between different frequency components in the pantograph-catenary system (PCS). This integrated approach stands in contrast to other strategies: local feature extraction methods like the self-supervised pre-training method [3] employ domain-specific masking to retain local structural features of railway components, achieving high efficiency in small object detection. Meanwhile, multi-modal methods such as [4] fuse local cues from heterogeneous sensors (e.g., infrared/visible) to resolve ambiguities in complex scenes. However, these methods lack explicit mechanisms for modeling global system dynamics.

2.Section 3.1 Network Architecture:The manuscript should provide a clear description of the inputs and outputs between modules. For Figure 1, it is suggested to use a network architecture diagram of DINO rather than a process flowchart for better clarity. Given that the code is not open-sourced, it is also advisable for the authors to provide a pseudo-algorithm to facilitate understanding and reproducibility.

Similarly, Figure 2 should indicate the input and output variables between the modules to enhance the comprehensibility of the network's structure.In the section describing the "Efficient Local-Global Fusion Transformer Algorithm", the manuscript should provide explicit details of the model's loss function and optimization process.

Answer: We thank the reviewer for this valuable suggestion.

(1)We have replaced the original process flowchart with a detailed network architecture diagram of DINO, which makes the architecture easier to interpret.

(2)We have added a concise pseudo-code representation of the main computational pipeline. This addition aims to facilitate replication of our work.

(3)We appreciate this valuable suggestion. Similar to DINO, we also use multiple CDN groups to improve the effectiveness of our method. The reconstruction losses are and GIOU losses for box regression and focal loss for classification. The loss to classify negative samples as background is also focal loss.

3.Experimental Section : Besides quantitative evidence, the authors are encouraged to provide qualitative evaluations to demonstrate the superiority of the proposed model, such as attention or heat maps to visualize the learned features and support the claims.

It is recommended to add a discussion subsection in the experimental part, analyzing scenarios where the model's performance degrades. The authors should discuss possible reasons for such degradation and outline potential directions for future work to address these limitations. Overall, this is an interesting and valuable study. However, substantial revisions are required before publication.

Answer: We sincerely thank the reviewer for their constructive feedback and positive evaluation of our work. In the revised manuscript, we have implemented the following improvements to strengthen the experimental section:

(1)To complement the quantitative results, we have added qualitative visualizations of the learned feature representations. Specifically, Grad-CAM is employed to generate attention visualizations in Fig. 13, revealing the spatial regions that the model attends to during inference. These visualizations provide intuitive evidence of the proposed model’s ability to capture both global context and fine-grained details, supporting the performance claims in the paper.

(2)We have introduced a new “Discussion” subsection at the end of the experimental section. In the revised manuscript, we have expanded discussion as:

Although the proposed model demonstrates competitive performance in both quantitative metrics and qualitative visualization, somre failure cases of our method appear in several scenarios, e.g., low-light scenes or severe object occlusion. For low-light, the reduction in image contrast and color fidelity limits the discriminative capacity of visual features, leading to suboptimal predictions. For severe object occlusion, partial visibility of key semantic regions reduces the effectiveness of local feature extraction, making the representation less reliable. We attribute these degradations to two main factors: First, Data distribution bias, as the training dataset contains fewer samples representing extreme lighting or heavy occlusion conditions. Second, Limitations in spatial dependency modeling, where the current mechanism may not fully capture long-range relationships or adapt to highly variable object appearances. At present, transfer leanring and progressive hierarchical attention mechanisms has applied to address these problems. It is an interesting direction to apply these paradigms into small object detection in the future.

Reviewer #2

1.In the Abstract section of the manuscript, the sentence "Transformer-based algorithms have exhibited remarkable performance" could be improved for clarity and specificity. A refined version of this sentence would be: "Transformer-based algorithms have demonstrated remarkable performance in the domain of computer vision tasks." This revision explicitly mentions the domain where the performance is notable, providing a clearer context for the reader and emphasizing the relevance of the research in computer vision. Additionally, please note that we have reviewed your submission and require you to remove all figures from within your manuscript file, leaving only the individual TIFF/EPS image files. These files will be automatically included in the reviewer’s PDF. This process ensures that the images are of high quality and properly formatted for the review process.

Answer: We sincerely thank the reviewer for the thoughtful suggestions and careful attention to detail. The following revisions and adjustments have been made accordingly:

(1)We appreciate the recommendation to improve clarity and specificity. Following the suggestion, the sentence “Transformer-based algorithms have exhibited remarkable performance” in the Abstract has been revised to: Transformer-based algorithms have demonstrated remarkable performance in the domain of computer vision tasks.

(2)We also appreciate the reminder regarding figure submission requirements. In accordance with the journal’s guidelines, we have provided the corresponding TIFF/EPS image files.

Reviewer #3

1.The authors have made commendable progress in designing a lightweight model tailored for effective detection of small objects.

The manuscript is generally well-structured and clearly presented. The visualizations, especially the detection outputs, are intuitive and support the claims made.

However, the ablation study section would benefit from a more in-depth technical explanation. Clarifying the rationale behind each component tested, along with a more rigorous discussion of performance variations, would enhance the overall scientific contribution.

The results are presented effectively. Nevertheless, for completeness and consistency, it is recommended that detection outcomes across all datasets be visualized similarly to Figure 13.

Answer: Thank you very much for your positive recognition of our efforts. We are glad that our lightweight model design for small object detection meets expectations and contributes meaningfully to the field. We also sincerely appreciate your encouraging comments on the clarity of our manuscript and the effectiveness of our visualizations. This motivates us to maintain high standards in both presentation and scientific rigor. The following revisions and adjustments have been made accordingly:

(1)We have expanded the ablation study section to provide a detailed explanation of the purpose. Additionally, we include a thorough analysis of the observed performance changes, discussing the technical reasons behind improvements or degradations to strengthen the scientific depth of this part. In the revised manuscript, revision part is: The core idea of the GFUM module is that the six-layer encoder is divided into three groups. For each group, the first layer aggregates information globally through high-level features to reduce computational load, while the second layer preserves details using low-level features, achieving a balance between multi-scale feature maintenance and computational efficiency. Moreover, the ELGFA moudle by pooling image features from the height, width, and global dimensions, effectively integrates both local and global contextual information to enhance the input feature image, aiming to accurately identify regions of interest. Furthermore, the CB module computes the average of feature vectors across layers and then rebroadcasts this global context information to each feature vector.

(2)we have added visualizations of detection results for all datasets following the style of Figure 13, as shown in Fig. 15 and Fig. 16.

---

## [Editor Report · Decision Letter 1]

3 Sep 2025

ELFT: efficient local-global fusion Transformer for small object detection

PONE-D-25-27252R1

Dear Dr. Li,

We’re pleased to inform you that your manuscript has been judged scientifically suitable for publication and will be formally accepted for publication once it meets all outstanding technical requirements.

Kind regards,

Aiqing Fang

Academic Editor

PLOS ONE

Additional Editor Comments (optional):

Strongly recommend the author to submit vector graphics for printing

---

## [Editor Report · Acceptance letter]

PONE-D-25-27252R1

PLOS ONE

Dear Dr. Li,

I'm pleased to inform you that your manuscript has been deemed suitable for publication in PLOS ONE. Congratulations! Your manuscript is now being handed over to our production team.

Kind regards,

on behalf of

Dr. Aiqing Fang

Academic Editor

PLOS ONE